# Quantum localization and delocalization of charge carriers in organic semiconducting crystals

Samuele Giannini [1], Antoine Carof [1], Matthew Ellis[1], Hui Yang[1], Orestis George Ziogos[1], Soumya Ghosh[1] & Jochen Blumberger[1,2]

Charge carrier transport in organic semiconductors is at the heart of many revolutionary technologies ranging from organic transistors, light-emitting diodes, flexible displays and photovoltaic cells. Yet, the nature of charge carriers and their transport mechanism in these materials is still unclear. Here we show that by solving the time-dependent electronic Schrödinger equation coupled to nuclear motion for eight organic molecular crystals, the excess charge carrier forms a polaron delocalized over up to 10–20 molecules in the most conductive crystals. The polaron propagates through the crystal by diffusive jumps over several lattice spacings at a time during which it expands more than twice its size. Computed values for polaron size and charge mobility are in excellent agreement with experimental estimates and correlate very well with the recently proposed transient localization theory.

[1] Department of Physics and Astronomy and Thomas Young Centre, University College London, London WC1E 6BT, UK. [2] Institute for Advanced Study, Technische Universität München, Lichtenbergstrasse 2 a, D-85748 Garching, Germany. Correspondence and requests for materials should be addressed to J.B. (email: j.blumberger@ucl.ac.uk)

Organic semiconductors (OSs) differ from inorganic semiconductors in two important aspects: they are made of small or polymeric molecules that are held together by weak van-der-Waals interactions rather than covalent bonds. Hence, thermal motions of the molecules around their lattice positions is very pronounced and leads to large fluctuations of electronic coupling, also termed off-diagonal electron–phonon coupling. Secondly, the static dielectric constant of OSs is typically very small and, as a consequence, the reorganization energy $\lambda$ or local electron–phonon coupling is small, too (0.2 eV or less). These two material properties place charge transport in OSs in a regime that challenges traditional transport descriptions[1–6]. The band-description asserts the existence of "Bloch" states and typically breaks down at ambient temperatures where the mean free path of the scattered carrier becomes smaller than the intermolecular lattice spacing[3,7,8]. The polaronic band models of Ortmann and Hannewald successfully reconciles some of the effects of coupled nuclear and electronic motion[9] but they too become problematic at ambient temperature[3]. Charge hopping models on the other hand assert the existence of a finite free energy barrier for charge hopping and a separation of time scales between charge transfer and the molecular motions coupled to it[3,5,6,10]. For common OSs like rubrene, pentacene and $C_{60}$ one or both requirements are not met[11,12]. In recent years, the community has pursued the development of either more advanced theories, e.g., transient localization theory[4,13–15], or approximate quantum dynamical direct propagation methods[16–20]. Yet, the latter were mostly limited to simple displaced harmonic oscillator model Hamiltonians and often the so-called back reaction from the electronic to the nuclear degrees of freedom was not accounted for.

What is needed to advance our understanding of charge transport in OSs are numerically efficient, reliable and practical direct propagation schemes for electron-nuclear motion that are free of limiting model assumptions, that seamlessly bridge the gap between different mechanistic regimes and that support assumptions of alternative transport theories[4,13–15]. Here we show that mixed quantum-classical non-adiabatic molecular dynamics in the framework of our recently developed fragment-orbital based surface hopping (FOB-SH) method[21–23] is a truly predictive approach in this regard, in particular at ambient and high temperature where nuclear quantum effects are still relatively small[24]. While previous investigations were limited to charge transport in short chains of small molecules[23], latest algorithmic developments now allow us to apply FOB-SH for the first time to charge transport in realistic nano-scale systems formed of up to a few hundred medium-sized organic molecules.

## Results

**Charge carrier transport mechanism in 2D materials**. In this work we use the FOB-SH methodology to uncover the nature and transport mechanism of charge carriers in eight single crystalline OSs, each exhibiting 2D conductance in their herringbone layers yet a significantly smaller or vanishing conductance in the respective orthogonal direction (see Fig. 1). The systems were chosen to represent low, medium and highly conductive OSs with experimental mobilities spanning three orders of magnitude. The electronic structure calculations of important transport parameters such as electronic couplings and reorganization energies, as well as the subsequent parametrization of the molecular model are described in Methods, with parameters summarized in Table 1. We note in passing that reorganization energy is assumed to be equal to the intramolecular (or "inner-sphere") contribution. The intermolecular (or "outer-sphere") contribution is typically very small in apolar OSs[25,26] studied here and is

neglected. Details on the FOB-SH method and simulation protocols as well as a discussion of important properties including detailed balance, internal consistency (Supplementary Fig. 1) and decoherence (Supplementary Fig. 2) are given in the Methods section.

The initial dynamics of the hole carrier wavefunction $\Psi(t)$ and the polaron size (defined by the inverse participation ratio (IPR) as described in Methods) over the first 100 fs are shown in Fig. 2 for two representative OSs ($T = 300$ K): panels (a)–(f) for the low mobility OS pMSB and panels (g)–(l) for the high mobility OS pentacene. Starting from an electronic wavefunction that is initially localized on a single pMSB molecule (Fig. 2b), we observe frequent hops of the electron hole, each involving rapid delocalization of the hole carrier wavefunction over a few molecules (Fig. 2c(e)) and re-localization on a single molecule that is one or a few molecular spacings (0.3–0.5 nm) apart (Fig. 2d(f)). The average IPR is equal to 1.7 and the root-mean-square fluctuation $\sigma$ equals 0.9 (see Table 2). The observed hole hopping mechanism is not unexpected for this OS because the thermal average of electronic coupling between the molecules, $V = \langle |H_{kl}|^2 \rangle^{1/2}$, is significantly smaller than reorganization energy, $\xi = 2V/\lambda = 0.1$. However, the mechanism differs from the Marcus picture often used to model small polaron hopping in OSs[6], in that several molecules bridging initial donor and final acceptor may come simultaneously into energetic resonance resulting in hole transfer to a molecule beyond the nearest neighbor in a single hopping event. This is more reminiscent of the flickering resonance mechanism recently proposed by Skourtis and Beratan for hole transport in DNA[27].

The situation is strikingly different for pentacene. The initially localized electronic wavefunction $\Psi(t)$ (Fig. 2h) rapidly spreads over many molecules (Fig. 2i) to form a polaron that is preferentially delocalized along the $T_1$ direction where $\pi$-orbital overlap and hence electronic coupling between neighboring molecules is the largest (Fig. 2j). On average, the polaron is delocalized over 18 molecules ($\sigma = 10.3$) in excellent agreement with estimates based on experimental electron spin resonance data, 17 molecules at 290 K[28]. Delocalization occurs because electronic coupling is now on the same order of magnitude as reorganization energy, $\xi = 2.2$, which brings several molecules simultaneously into energetic resonance at any point in time. Yet, disorder in the site energies and electronic couplings prevent the wavefunction from further delocalization. In FOB-SH this effect is born out by the wavefunction $\Psi(t)$ projecting on the ground or low energy electron hole eigenstates (i.e., states close to the valence band edge, see Supplementary Fig. 1), which are delocalized over no more than a dozen molecules. The motion of the polaron within the herringbone layer of pentacene is particularly intriguing. Neighboring clusters of molecules frequently come into energetic resonance with the polaron causing $\Psi(t)$ to expand to about twice its size for short durations of time (Fig. 2k). At this point $\Psi(t)$ projects on higher-lying electron hole eigenstates (i.e., states closer to the middle of the valence band), which are more extensively delocalized, typically over 20–50 molecules. Some of these sudden bursts of the wavefunction are successful, meaning $\Psi(t)$ returns to a low-lying electron hole eigenstate that is localized on a neighboring cluster of molecules (Fig. 2l).

The dynamics at longer times, up to a few picoseconds, is shown in Fig. 3 for both materials. We find that the average duration of a "resonance", defined here by the time it takes for the IPR to exceed and subsequently return below $\langle IPR \rangle + \sigma$ is 7 and 12 fs for pMSB and pentacene, respectively, see Fig. 3a and b, which is close to the characteristic oscillation time of intramolecular vibrations and site energy fluctuations. The average time between two resonances is about an order of magnitude larger,

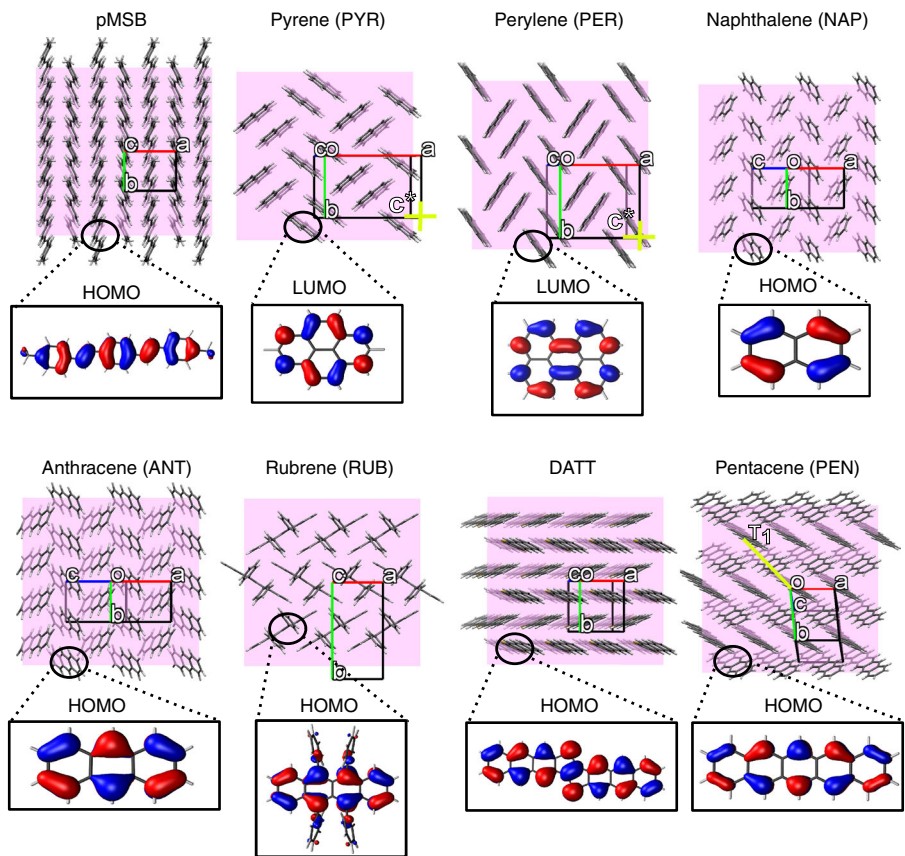

**Fig. 1** Molecular herringbone layer packing for all investigated OSs. The unit cell axes *a*, *b*, *c* are shown in red, green and blue, the herringbone layer is in the *a*-*b* plane, other specific directions discussed in the main text are shown in yellow. The DFT highest occupied molecular orbital (HOMO) and lowest unoccupied molecular orbital (LUMO) of single molecules are depicted as isosurfaces for OSs where hole transfer and electron transfer is studied, respectively. See Supplementary Table 1 for references to the experimental crystal structures shown

**Table 1 Computed transport parameters for the OSs in this work**

| Crystal[a] | dir. | dist. (Å) | $H_{kl}$(FODFT)[b] (meV) | $H_{kl}$(AOM)[c] (meV) | $V$[d] (meV) | $\sigma_V$[e] (meV) | $\lambda$[f] (meV) | $\sigma_{\Delta E}$[g] (meV) |
|---|---|---|---|---|---|---|---|---|
| DATT-h$^+$ | *a* | 6.26 | 94.9 | 74.8 | 76.1 | 22.7 | 88.0 | 67.3 |
| RUB-h$^+$ | *a* | 7.18 | 113.4 | 111.9 | 101.8 | 33.5 | 152.0 | 91.2 |
| PEN-h$^+$ | $T_1$ | 4.80 | 116.1 | 124.6 | 110.8 | 31.1 | 98.0 | 73.8 |
| ANT-h$^+$ | *a* | 5.24[h] | 17.6 | 30.7 | 29.6 | 30.6 | 142.0 | 87.7 |
| | *b* | 6.04 | 57.2 | 57.4 | 51.6 | 27.2 | 142.0 | 89.5 |
| NAP-h$^+$ | *b* | 5.95 | 46.2 | 41.4 | 35.9 | 19.4 | 187.0 | 103.0 |
| PER-e$^{-f}$ | *a* | 6.10[h] | 61.7 | 52.6 | 41.6 | 16.6 | 177.0 | 101.1 |
| | *c** | 10.26 | 8.3 | 7.0 | 10.4 | 6.5 | 177.0 | 100.2 |
| PYR-e$^-$ | *c** | 8.47 | 26.7 | 18.7 | 18.0 | 14.6 | 222.0 | 105.8 |
| pMSB-h$^+$ | *b* | 5.88 | 21.5 | 25.2 | 17.2 | 8.6 | 254.6 | 113.3 |

[a]Reference to the crystal structures used in this work are given in Supplementary Table 1
[b]Electronic couplings for crystal structure geometries obtained using scaled FODFT as described in Molecular model section. Comparison between FODFT and literature values is given in Supplementary Table 1
[c]Electronic couplings for crystal structure geometries, $H_{kl} = C_{lin}\bar{S}_{kl}$, $C_{lin}$ from Supplementary Fig. 5
[d]Mean electronic couplings averaged over MD trajectories, $V = \langle|H_{kl}|^2\rangle^{1/2}$
[e]Fluctuations of electronic couplings from MD trajectories, $\sigma_V = \sqrt{\langle|H_{kl}|^2\rangle - \langle|H_{kl}|\rangle^2}$
[f]Reorganization energy (using 4-points calculation as detailed in Methods)
[g]Site energy fluctuations, $\sigma_{\Delta E} = \sqrt{\langle\Delta E_{kl}^2\rangle - \langle\Delta E_{kl}\rangle^2}$
[h]T-shaped molecular pair along the given direction

52 fs for pMSB and 114 fs for pentacene. Similar values are obtained for the other compounds, see Table 2. These resonances give rise to spatial displacements as described qualitatively above and shown in Fig. 3 by way of projecting $\Psi(t)$ on the crystallographic directions *b* and $T_1$ of pMSB (Fig. 3c) and pentacene (Fig. 3d), respectively. Yet, significant displacements along these directions occur at somewhat longer times than the time between two resonances, more characteristic of the oscillation time of the electronic coupling fluctuations, $\tau = 159$ and 202 fs rad$^{-1}$ for pMSB and pentacene, respectively, see Fig. 3c and d. Hence, as one would expect, only a fraction of the resonances (estimated to be about 0.2–0.5) leads to a successful

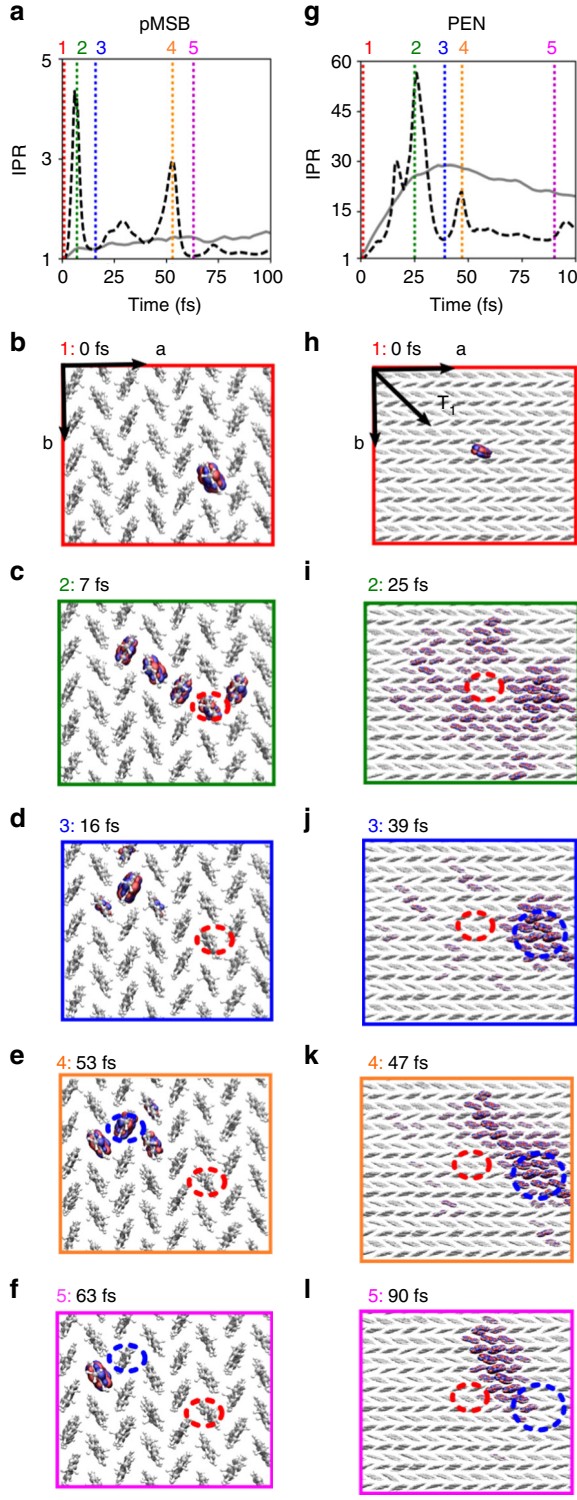

**Fig. 2** Time evolution of the charge carrier wavefunction in the first 100 fs. The number of molecules over which the polaron is delocalized, defined by the inverse participation ratio (IPR) (see Methods) is shown in (**a**, **g**) for pMSB and pentacene, respectively, against time. Black dashed lines are used to indicate representative single FOB-SH trajectories and gray solid lines are averages over 300 trajectories. In (**b–f**) and (**h–l**) snapshots of the hole carrier wavefunction $\Psi(t)$ (see definition in Methods) in the respective herringbone layers are shown starting from a fully localized wavefunction at time t = 0. The snapshots are taken from the same single trajectories in (**a**, **g**) at the times indicated by vertical dotted lines with different colors. Isosurfaces of the magnitude of the wavefunction, $|\Psi(t)| = 2 \times 10^{-3}$, are shown and colored according to the phase $\theta$, $\Psi(t) = |\Psi(t)|\exp(i\theta)$: $-\pi/4 \leq \theta \leq 3\pi/4$ in blue and $3\pi/4 < \theta < 7\pi/4$ in red. Only a zoomed-in region of the simulated herringbone layer is shown and the molecules perpendicular to the herringbone layer are removed to enhance visibility

**Table 2 Characterization of polaron size (IPR) and its thermal fluctuations[a]**

| crystal | $\langle IPR \rangle$[b] | $\sigma$[c] (IPR) | $t_r^d$ (fs) | $\tau_r^e$ (fs) | $\tau^f$ (fs rad$^{-1}$) |
|---|---|---|---|---|---|
| DATT-h$^+$ | 15.9 | 10.8 | 15 | 133 | 159 |
| RUB-h$^+$ | 13.7 | 8.2 | 9 | 71 | 333 |
| PEN-h$^+$ | 17.4 | 10.3 | 12 | 114 | 202 |
| ANT-h$^+$ | 4.9 | 2.6 | 9 | 73 | 398 |
| NAP-h$^+$ | 2.5 | 1.4 | 8 | 58 | 114 |
| PER-e$^-$ | 3.3 | 1.6 | 9 | 87 | 199 |
| PER-e$^-$-c* | 1.1 | 0.1 | 12 | 277 | – |
| PYR-e$^-$-c* | 1.2 | 0.3 | 9 | 164 | – |
| pMSB-h$^+$ | 1.7 | 0.9 | 7 | 52 | 159 |

[a]All values are averaged over 600 FOB-SH trajectories of approximate length 1 ps. The first 200 fs of dynamics were discarded
[b]Average of IPR
[c]Root-mean-square fluctuations of IPR
[d]Average duration of a resonance. The duration of a resonance is defined by the time it takes for the IPR to exceed and subsequently return below $\langle IPR \rangle + \sigma$
[e]Average time between two resonances
[f]Characteristic oscillation time of electronic coupling, corresponding to the peak of highest intensity at $\omega_0$ in the power spectrum of electronic coupling fluctuations from a MD trajectory (5 ps long), $\tau = \omega_0^{-1}$, where $\omega_0$ is the angular frequency

displacement. Notably, the wavefunction displacements in pentacene are over several lattice spacings at a time, 3–5 nm, that is about an order of magnitude larger than the (mostly nearest-neighbor) displacements in pMSB. As we will see in the following, this difference gives rise to a ≈50-fold higher charge mobility in pentacene relative to pMSB.

**Charge mobility and wavefunction delocalization.** For the calculation of charge mobility we run 1000 FOB-SH trajectories for

each system to obtain the mean-square displacement (MSD) of the center of $\Psi(t)$ as a function of time (see Methods section). After a short initial relaxation period we observe a linear increase of the MSD with time, implying that the Einstein diffusion approximation is valid (Supplementary Fig. 3). The charge mobilities obtained from the Einstein relation are shown in Fig. 4a (data in blue). They are in excellent agreement with experiments or within the experimental error bars where uncertain, with typical deviations of less than a factor of two for mobilities spanning 3 orders of magnitude. We find that charge mobility correlates very well with both: average polaron size, as defined by the inverse participation ratio (IPR) (Fig. 4b), and the order parameter $\xi$ (Fig. 4c) determining the existence and height of the free energy barrier for charge transfer between nearest neighbors, as illustrated in Fig. 4d. As discussed below, traditional hopping and band models fail to provide a uniformly good description of charge transport in the OSs investigated.

FOB-SH mobilities up to ≈1 cm$^2$ V$^{-1}$ s$^{-1}$ including the one for pMSB ($\xi < 0.2$), are well reproduced by a chemical master equation for small polaron hopping between nearest neighbors with hopping rates from electron transfer (ET) theory, as described in Methods, (data in dark green in Fig. 4a), despite our observation above for pMSB that the actual mechanism is more intricate than simple nearest neighbor hopping. For OSs

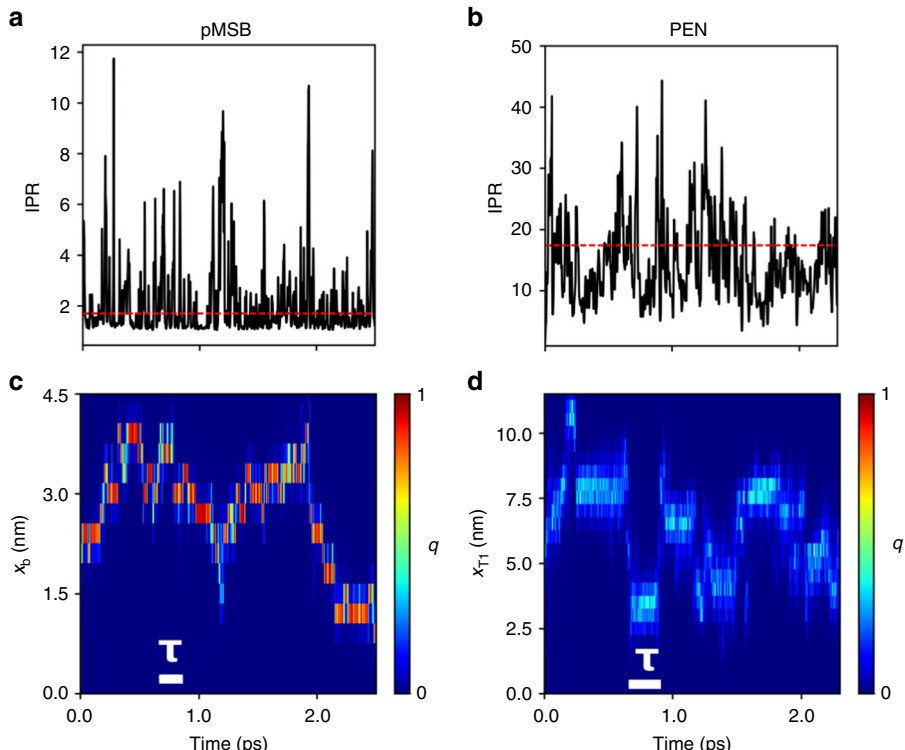

**Fig. 3** Time evolutions of IPR and carrier wavefunction on the picosecond time scale. A single representative FOB-SH trajectory at $T = 300$ K in the herringbone layer of pMSB (**a**, **c**) and pentacene (**b**, **d**) is illustrated. In **a**, **b**, the IPR is reported with black lines and the average IPR, given in Table 2, with dashed red lines. In **c** the quantum amplitudes of the molecules within the herringbone layer, $|u_i(t)|^2$, are projected on the $b$ direction, $q(x_b, t) = \sum_{i, x_{b,i}=x_b}^{\text{molecules}} |u_i(t)|^2$, and in (**d**) the projection is on the $T_1$ direction, $q(x_{T1}, t) = \sum_{i, x_{T1,i}=x_{T1}}^{\text{molecules}} |u_i(t)|^2$. The charge carrier is strongly localized in regions colored in red and delocalized in regions colored in light blue. The time scale characteristic for electronic coupling fluctuations, $\tau$ (see Table 2), is indicated by white bars. At $t = 0$, $\Psi(t)$ is fully localized on a single molecule ($q = 1$) in both materials. In pMSB small polaron hopping events (motion along $x_b$) and in pentacene large diffuse jumps of a delocalized polaron are observed (motion along $x_{T_1}$)

with larger mobilities, $\approx 1-5$ cm$^2$ V$^{-1}$ s$^{-1}$ ($0.2 < \xi < 1$), the free energy barrier is small, causing the polaron to delocalize over 2–5 molecules according to FOB-SH simulations. Hence, in this regime, the small polaron hopping model assuming nearest neighbor hops of a fully localized charge carrier is no longer a good physical model of the charge transport process. Nonetheless, if one solves the chemical Master equation with nearest neighbor hopping rates from ET theory, the resultant mobilities are in good agreement with FOB-SH and experimental values (data in shaded green). This agreement appears to be coincidental as the small polaron hopping mechanism bears no resemblance with the actual mechanism obtained from FOB-SH. Indeed, it is well known that a small polaron hopping model may give the same order of magnitude in mobility or current as a larger polaron model[29] - agreement with the experimental mobility gives no sufficient information on the mechanism.

At even higher mobilities, $\gtrsim 5$ cm$^2$ V$^{-1}$ s$^{-1}$ ($\xi \geq 1$), the free energy barrier disappears completely and polarons are delocalized over several to many molecules, as observed above for pentacene. In this regime band theory does not give an adequate description either: experimental mobilities are overestimated due to strong thermal motions violating basic assumptions of this theory (data in shaded red; only for still higher mobilities this theory becomes valid). By contrast, FOB-SH describes all regimes relevant to OSs accurately and seamlessly bridges the gap between small polaron hopping and band transport.

Our results obtained from explicit time propagation of the electron-nuclear dynamics can be used to test more recent

theoretical models of charge transport in OSs, e.g., the transient localization theory (TLT) proposed by Fratini and Ciuchi[4,13–15]. This theory is based on the observation that electronic coupling fluctuations on the time scale $\tau = 0.1 - 1$ ps rad$^{-1}$, cause a transient localization of the charge carrier, in agreement with what we observe for pentacene in Fig. 3b. The main result of TLT is that the mobility is related to the squared transient localization length of the carrier wavefunction, $L_\tau^2$, $\mu_{\text{TLT}} = eL_\tau^2/(2k_B T \tau)$, where $e$ is the unit charge, $k_B$ the Boltzmann constant and $T$ the temperature. We have calculated $\mu_{\text{TLT}}$ from $L_\tau^2$ using the electronic Hamiltonians sampled along present FOB-SH trajectories and setting the site energies to zero (see Table 2 for values of $\tau$). We find that TLT gives indeed a good prediction of experimental values in the high mobility regime (see Fig. 5a, data in green). If site energy fluctuations are retained in the electronic Hamiltonian, TLT also captures the hopping-like regime, albeit concomitant with a slight increase in deviation for the high mobility regime (data in red). We find also a good correlation between our IPR and the localization length $L_\tau^2$ divided by the area per molecule within the herringbone layer, as shown in Fig. 5b.

**Charge mobility limiting factors.** An important objective in the discovery process of efficient OSs is the understanding of the aspects limiting polaron delocalization and ultimately charge mobility. According to Troisi and co-workers the major limiting factor are the thermal fluctuations of electronic coupling between the molecules leading to localization of the electronic

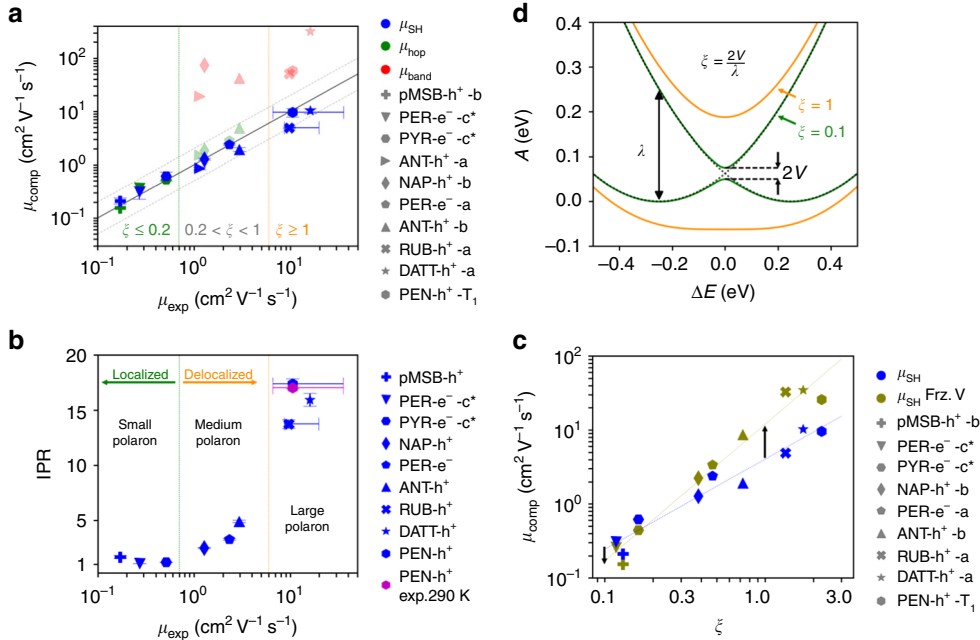

**Fig. 4 Charge mobility and IPR for all investigated OSs. a** Computed versus experimental charge mobilities for the OS materials shown in Fig. 1; pMSB-h$^+$-b denotes hole transport along the $b$ crystallographic direction, and a similar notation is used for the other systems. Charge mobilities from FOB-SH (data in blue) are obtained by averaging the MSD of the charge carrier wavefunction $\Psi(t)$ over 1000 trajectories and inserting the corresponding diffusion coefficients in the Einstein relation (see Methods for detailed equations and Supplementary Fig. 3 for MSD used). Statistical error bars indicate the standard deviations over five independent blocks of 200 trajectories. They are small and may be invisible. Experimental error bars for RUB and PEN are based on the measurements cited in Table 3. Predictions from band theory calculations are taken from the literature (data in red, see Table 3 for references). Charge mobilities from a small polaron hopping model (data in green) are obtained by solving a chemical Master equation for nearest neighbor hopping in the specified direction using semi-classical ET rates (see Methods). As a guide to the eye, perfect agreement is indicated by a thick solid line and deviations in mobility by a factor of 2 by thin dotted lines. **b** Correlation between time-averaged IPR and measured mobilities. The experimental estimate for the size of the hole polaron in pentacene was taken from ref. [28]. Error bars were obtained by block averaging the equilibrated region of the IPR. **c** Influence of the thermal fluctuations of electronic coupling (off-diagonal electron–phonon coupling) on charge mobility. Data in olive are obtained from FOB-SH with electronic coupling between the molecules frozen to the thermal average. Data in blue are taken from (**a**) and shown for comparison. Note the significant increase in charge mobility for systems forming large polarons. **d** Diabatic (black dashed) and adiabatic free energy profiles (solid) for electron transfer between a donor and an acceptor, defining reorganization free energy, $\lambda$, average electronic coupling, $V$, and the parameter $\xi$ determining existence and height of the barrier

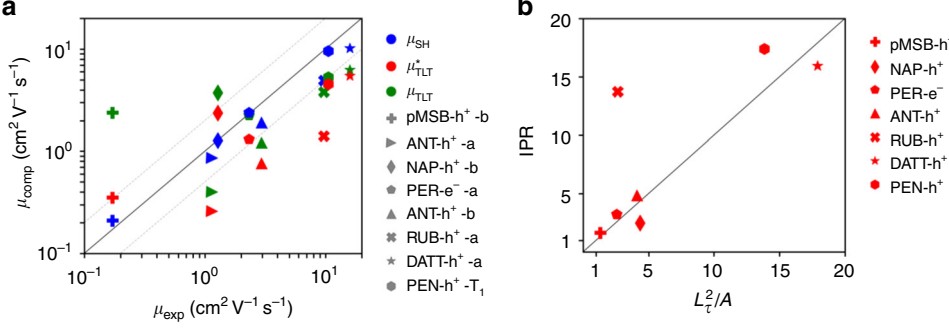

**Fig. 5 Comparison between FOB-SH and transient localization theory. a** Charge mobilities obtained from TLT vs experiment for the OS materials shown in Fig. 1. The squared transient localization length $L_\tau^2$ is calculated, as described in ref. [15] using electronic Hamiltonians from present FOB-SH trajectories, firstly, without modification of the onsite energies and their thermal fluctuations (data in red) and, in addition, after removal of onsite energy fluctuations by setting all diagonal matrix elements to zero (data in green), see Methods for details. Values for $\tau$ are taken from Table 2. FOB-SH mobilities in blue are taken from Fig. 4a and shown for comparison. **b** Correlation between IPR and $L_\tau^2/A$, where $A$ is the area per molecule within the herringbone layer

eigenstates and hence to reduction in mobility[3,13,14,16]. Indeed several attempts have already been made experimentally to reduce off-diagonal disorder, with some successes[30,31]. To estimate the maximum possible boost in charge mobility that one could achieve via complete removal of off-diagonal electron–phonon coupling, we carried out FOB-SH simulations with electronic couplings frozen to their mean values (Fig.

4c, data in olive). While in the small polaron hopping regime ($\xi < 0.2$) the mobility slightly decreases, as one would expect from non-adiabatic ET theory, in the medium and large polaron regime ($\xi > 0.2$) the mobility increases significantly, by up to a factor of 7 for rubrene. Yet, the charge carrier is still polaronic due to the thermal fluctuations of the site energies (diagonal electron-phonon coupling). If the latter are frozen as

well, the polaron fully delocalizes and the band transport regime is reached.

## Discussion

In conclusion, we have demonstrated that FOB-SH enables unprecedented insight into the elusive, intricate and long-debated nature and dynamics of charge carrier transport in crystalline OSs, based on rigorous physical principles. In contrast to traditional transport theories, it provides a sound description of the notoriously challenging but practically important charge transport regime of room temperature high-mobility OS materials. Although FOB-SH is a non-adiabatic molecular dynamics (MD) method, it is fairly computationally inexpensive, with a cost per MD step that is typically 2–35 times higher than that for a classical molecular dynamics simulation on systems with a few hundreds to a thousand molecules, respectively. Therefore we expect this methodology to become a practical tool for the computer-aided design of next-generation high-mobility OS materials, and more generally for the realistic prediction of charge transport mechanisms in "soft" condensed matter including wet biological molecules.

## Methods

**Fragment orbital-based surface hopping (FOB-SH).** FOB-SH is a mixed quantum-classical fewest switches surface hopping technique that permits efficient simulation of charge and exciton transport in condensed phase materials[21–23,32]. In FOB-SH, it is assumed that the complicated many-body dynamics of an excess electron or electron hole can be effectively described by a time-dependent one-particle wavefunction, $\Psi(t)$. The latter is expanded in a basis of fragment or site orbitals, here the frontier orbitals of the charge mediating molecules, that is, the highest occupied molecular orbitals (HOMO) for hole transport or the lowest unoccupied molecular orbitals (LUMO) for electron transport,

$$\Psi(t) = \sum_{l=1}^{M} u_l(t)\phi_l(\mathbf{R}(t)), \quad (1)$$

where $\mathbf{R}(t)$ denotes the time-dependent nuclear positions. The wavefunction is propagated according to the electronic Schrödinger equation in the time-dependent potential due to nuclear motion,

$$i\hbar\dot{u}_k(t) = \sum_{l=1}^{M} u_l(t)[H_{kl}(\mathbf{R}(t)) - i\hbar d_{kl}(\mathbf{R}(t))], \quad (2)$$

where $H_{kl} = \langle\phi_k|H|\phi_l\rangle$ and $d_{kl} = \langle\phi_k|\dot{\phi}_l\rangle$ are the electronic Hamiltonian matrix elements and non-adiabatic coupling elements (NACEs) in the (quasi-diabatic) site orbital basis $\{\phi_l\}$. The diagonal matrix element of the electronic Hamiltonian or site energy, $H_{kk}$, is the total electronic energy of the system when the charge carrier is localized on molecule $k$ while all other molecules $k \neq l$ are charge neutral. The off-diagonal matrix elements, $H_{kl}$, are often referred to as electronic couplings or transfer integrals. Both, site energies and electronic couplings fluctuate due to nuclear motion (note dependence on $\mathbf{R}(t)$ on the right hand side of Eq. (2)), which is referred to as diagonal and off-diagonal electron-phonon coupling. The nuclei propagate on a single ("active") adiabatic electronic potential energy surface at any time (as obtained by diagonalization of $H_{kl}$) and hop stochastically between different surfaces according to Tully's hopping probability[33] (in the current context, not to be confused with the charge carrier hopping mechanism). Trivial or non-avoided crossings between the dense adiabatic PESs in our systems are dealt with using a recently implemented state tracking algorithm[23]. The electronic decoherence is corrected by exponential damping of all except the active adiabatic electronic states using the Heisenberg principle-based decoherence time[22,23]. The decoherence correction occasionally leads to artificial long-range charge transfer that is removed with a projection algorithm as detailed in ref. [23].

A key feature of FOB-SH is that explicit electronic structure calculations of the elements $H_{kl}$ and their nuclear derivatives are avoided during time propagation, which allows us to investigate large systems and long time scales. The site energies $H_{kk}$ and gradients $\nabla_{\mathbf{R}}H_{kk}$ are approximated with a classical force field, while electronic couplings $H_{kl}$, coupling derivatives $\nabla_{\mathbf{R}}H_{kl}$ and NACEs $d_{kl}$ between the site orbitals are calculated using the analytic overlap method (AOM)[34]. The site energies and AOM couplings are parametrized from DFT calculations, see section Molecular model below. The electronic Hamiltonian and the nuclear derivatives are calculated every MD time step in the site basis and transformed to the adiabatic basis for propagation of the nuclei and calculation of the hopping probabilities. For further details of the method we refer to our recent publications[21–23].

**Molecular model.** Each molecule of the simulated systems can exist in two charge states: neutral and charged. The intra- and inter-molecular interaction terms for the neutral state are taken from the Generalized Amber Force Field (GAFF)[35]. For the charged state, the equilibrium bond lengths of the molecule are displaced with respect to the neutral state so that the reorganization energy $\lambda$ obtained from the force field is equal to the value obtained from DFT calculations,

$$\lambda = [E_C(\mathbf{R}_N) + E_N(\mathbf{R}_C)] - [E_C(\mathbf{R}_C) + E_N(\mathbf{R}_N)] \quad (3)$$

where $E_{C(N)}(\mathbf{R}_{N(C)})$ is the energy of the charged (neutral) molecule in the optimized neutral (charged) state and $E_{C(N)}(\mathbf{R}_{C(N)})$ is the energy of charged (neutral) molecule in the optimized charged (neutral) minimum. The geometry of charged and neutral molecules were optimized with the B3LYP functional and the 6–311g(d) basis set using the Gaussian program[36]. The reorganization energies obtained are summarized in Table 1 and the displacement of equilibrium bond lengths are shown in Supplementary Fig. 4. Hybrid functionals are known to give good equilibrium structures and better energies for bond stretching than GGA functionals, which is important for the calculation of reorganization energies[11,37]. Taking anthracene as example, we obtain similar values for two of the most popular hybrid functionals, $\lambda = 142.1$ meV for B3LYP and 149.9 meV for PBE0, but a smaller value for the GGA functional BLYP, 102.4 meV, due to the well known deficiency of the latter functional to underestimate the energy for bond stretching. The results are well converged with respect to the basis set used. Only very small changes in $\lambda$ are obtained as the basis set is increased: 138.1, 142.1, and 138.5 meV for the 6–31G(d), 6–311G(d) and 6–311G+(d, p) basis sets, respectively, using the B3LYP functional.

The force field equilibrium bond lengths of the charged state was adjusted by scaling the DFT displacements until force field and DFT reorganization energy matched. The scaling constant $\beta$, summarized in Supplementary Fig. 4, is close to unity for all systems, which means that the displacements in the force field and in DFT are almost identical. All other intra- and inter-molecular parameters were chosen to be the same as for the neutral state. The site energies $H_{kk}$ and nuclear gradients $\nabla_{\mathbf{R}}H_{kk}$ are obtained by assigning molecule $k$ the force field parameters for the charged state and all other molecules $l \neq k$ the parameters for the neutral state. For the systems investigated, electrostatic interactions in the form of fixed point charges do not significantly alter the energetics of the charge localized states because only the charged molecule carries a net charge while the other molecules are charge neutral and apolar. Hence, for the purpose of computational efficiency, electrostatic interactions were switched off. Therefore, the very small contribution in reorganization energy due to intermolecular modes[6] (also termed outer-sphere reorganization in the chemistry literature) is neglected. We expect that this is no longer a good approximation for crystals formed of polar or hydrogen bonded molecules. In this case the full electrostatics including electronic polarization of the molecules should be included, as it is well known that site energy fluctuations and hence reorganization free energies are overestimated for fixed point-charge models[38,39].

The electronic coupling matrix elements $H_{kl}$, $k \neq l$, are calculated using AOM[34]. The first step involves the calculations of reference electronic couplings. They are obtained from FODFT[25,39,40] calculations on a set of molecular dimer geometries that is comprised of all nearest neighbor dimers in the crystal structure and in selected structures obtained from molecular dynamics simulation of the crystal at room temperature. The FODFT calculations are carried out with the CPMD program package[41] using the PBE exchange correlation functional. Core electrons are described by Troullier–Martins pseudo potentials, and the valence electron states are expanded in plane waves with a reciprocal space plane wave cutoff of 90 Ry. The dimers are centered in the simulation box, and a vacuum of 4 Å was applied in each dimension. Using the same functional and basis set, the accuracy of FODFT couplings was benchmarked before on the HAB11[42] and HAB7-databases[43] for electronic coupling for hole and electron transfer in π-conjugated organic dimers. While the mean relative unsigned error with respect to high-level ab-initio reference values was found to be reasonably small (27.9%), the values were slightly but uniformly underestimated. Hence, as recommended in the previous studies, we scaled the FODFT couplings for hole and electron transfer systems by a factor of 1.348 and 1.325 to obtain best estimates.

In the second step for calculation of $H_{kl}$, the DFT molecular frontier orbital (HOMO for hole transfer, LUMO for electron transfer) is projected on a minimum Slater basis of $p$ orbitals with optimized Slater decay coefficients taken from ref. [34] (completeness of projection ≥ 0.98). In this minimum Slater basis the overlap between the HOMO (LUMO) orbitals of two monomers forming a dimer, $\bar{S}_{kl}$, can be calculated analytically and is extremely fast due to the small number of basis functions involved. For π-conjugated systems, it is usually sufficient to include only one optimized Slater $p$–orbital per atom contributing to π–conjugation, in this case:

$$\bar{S}_{kl} = \sum_{i}^{\text{atoms}} \sum_{j}^{\text{atoms}} c_{p\pi,i} c_{p\pi,j} \langle p_{\pi,i}|p_{\pi,j}\rangle \quad (4)$$

where $i$ and $j$ run over all π–conjugated atoms in molecules $k$ and $l$, respectively, and $p_{\pi,i}$ is the Slater type orbital $p$ on atom $i$, $c_{p\pi,i}$ is the corresponding expansion coefficient obtained by projection of the DFT molecular frontier orbital. Importantly, we find good linear correlation between $\bar{S}_{kl}$ from Eq. (4) and $H_{kl}$ from FODFT, see Supplementary Fig. 5, which allows us to estimate $H_{kl}$ from $\bar{S}_{kl}$ very rapidly for any geometry sampled along the trajectories. We fit a simple linear function for each OS, $H_{kl} = C\bar{S}_{kl}$, which we refer to as AOM couplings, and $C$ is a

constant of proportion. The fits of the scaling factor $C$ in Supplementary Fig. 5 are done either by minimization of residuals of $\log(H_{kl})$ to weight the error of couplings over all orders of magnitude uniformly (giving $C = C_{\log}$), or by minimization of residuals of $H_{kl}$ to weight more strongly the error of the largest couplings, which determine mobility (giving $C = C_{\lin}$). In most systems both methods give very similar results with mean relative unsigned errors of 36% ($C_{\log}$) and 44% ($C_{\lin}$) for AOM couplings with respect to FODFT couplings (average error for all systems in Supplementary Fig. 5). The largest difference between $C_{\log}$ and $C_{\lin}$, obtained for pMSB-h$^+$, still results in a rather small uncertainty in the non-adiabatic ET rate ($k_{ET} \propto H_{kl}^2$), of a factor of $C_{\lin}^2/C_{\log}^2 = 2.5$. Each MD time step the HOMO (LUMO) on each molecule is updated, as described in detail in ref. [21] and $H_{kl}$ between molecular pairs is estimated from $\bar{S}_{kl}$ via the above linear relationship. The nuclear derivatives $\nabla_\mathbf{R} H_{kl}$ and the NACE $\bar{d}_{kl}$ are obtained from finite differences of the overlap Eq. (4) with respect to nuclear displacements and time increments, respectively[21].

Possible shortcomings of AOM arise from the fact that, although atomic orbitals comprising the HOMO (LUMO) follow the motion of the atoms during the dynamics, the expansion coefficients in Eq. (4) are frozen otherwise. However, our checks indicated that this is a very good approximation, especially for rigid molecules, where orbitals are stable against intermolecular vibrations. More sophisticated interpolation schemes, or machine learning techniques could be used in future to improve reconstruction of the orbitals along the dynamics. Another source of inaccuracy could be the minimal basis set employed in Eq. (4), where only a single orbital per atom is considered. Although the validity of this approximation has been successfully tested before[34], one could use a larger basis set to improve the orbitals representation.

**Simulation details**. Starting from the experimental crystal structures, we built the following supercells for each OS, with number of molecules per supercell included in parenthesis: $14 \times 14 \times 2$ (784) for pMSB; $2 \times 2 \times 28$ (448) for PYR; $10 \times 10 \times 2$ (800) for PER; $12 \times 16 \times 4$ (1536) for NAP; $18 \times 28 \times 2$ (2016) for ANT; $30 \times 15 \times 1$ (1800) for RUB; $25 \times 20 \times 2$ (2000) for DATT; $20 \times 30 \times 2$ (2400) for PEN. Each supercell was equilibrated to a target temperature of 300 K running NVT molecular dynamics simulation for 0.5 ns in a configuration where a single molecule $i$ is in the charged state and all other molecules neutral. The last configuration is used to run an additional 0.5 ns NVE trajectory on the same state to sample initial positions and velocities for the following FOB-SH simulations. A subset of the molecules within the herringbone layer ($a$-$b$ plane) or orthogonal to it ($c^*$ direction) containing molecule $i$ (as specified in more detail further below) was treated as electronically active, i.e., as "sites" for construction of the electronic Hamiltonian, with their frontier orbital (HOMO or LUMO) contributing to the expansion of the carrier wavefunction Eq. (1). All other molecules of the respective supercell were treated electronically inactive and interacted with the active region only via non-bonded interactions. The charge carrier wavefunction was initialized as the frontier orbital localized on molecule $i$, $\Psi(0) = \phi_i(0)$. In general, $\phi_i(0)$ is a linear combination of the adiabatic electronic states (i.e., electronic eigenstates) $\psi_j$, unless $\lambda$ is sufficiently large so that $\phi_i(0) = \psi_0(0)$; hence the active adiabatic potential energy surface on which the nuclei initially propagate was chosen randomly with a probability proportional to $|\langle \psi_j(0)|\Psi(0)\rangle|^2$. The electronic Schrödinger equation (Eq. (2)) was integrated using the Runge-Kutta 4th order algorithm and the nuclei were propagated using the velocity Verlet algorithm. The electronic time step was set to be one fifth of the MD time step. The latter is equal to 0.1 fs, as optimized before for similar $\pi$-conjugated systems[22], except for pMSB-h$^+$, PER-e$^-$-c$^*$ and PYR-e$^-$-c$^*$, where a MD time step of 0.05 fs was used. Every MD time step the surface hopping probability and the non-adiabatic coupling vectors (NACVs) are calculated and after a successful hop the velocity component parallel to the NACV is rescaled to conserve total energy[22]. After an unsuccessful ("frustrated") hop the sign of the velocity component parallel to the NACV was inverted following Tully's prescription[44] which was found to slightly improve internal consistency[22]. State-tracking for detection of trivial crossings, decoherence correction and a projection algorithm for removal of decoherence correction-induced artificial long-range charge transfer were applied as described in ref. [23]. All surface hopping simulations were carried out in the NVE ensemble using our in-house implementation of FOB-SH in the CP2K simulation package[45].

**Calculation of IPR**. The carrier wavefunction $\Psi(t)$ was used to calculate the two main observables in this work, polaron size defined by the inverse participation ratio (IPR), and charge mobility, $\mu_{SH}$. The IPR is a common measure for the number of molecules over which the carrier wavefunction is delocalized[12,14,17,20],

$$\mathrm{IPR}(t) = \frac{1}{N_{traj}} \sum_{n=1}^{N_{traj}} \frac{1}{\sum_{\nu=1}^{N_{mol}} |u_{\nu,n}(t)|^4}. \tag{5}$$

where $u_{\nu,n}(t)$ is the expansion coefficient for the site orbital on molecule $\nu$ in trajectory $n$, $N_{mol}$ is the total number of electronically active molecules and $N_{traj}$ the number of FOB-SH trajectories. At first we investigated the convergence of the time-averaged IPR with respect to the number of electronically active molecules within the herringbone layer. While for low mobility OSs a few dozens of electronically active molecules are sufficient to converge the IPR, for medium and high

mobility OSs a few hundred molecules within a herringbone layer are required. The convergence for NAPH, PER, ANT, RUB, DATT, and PEN are shown in Supplementary Fig. 6, where each data point is an average over at least 200 FOB-SH trajectories of length 1 ps. On this basis, we chose for calculation of the IPR in Figs. 2 and 4 a square-shaped region of the herringbone layer containing the following number of electronically active molecules: 112 for pMSB; 315 for PYR; 238 for NAP; 323 for ANT; 783 for RUB; 888 for DATT; 900 for PEN. For PYR and PER in the orthogonal c$^*$ directions a 1D chain was selected as detailed below. For each system 600 FOB-SH trajectories of length 1 ps were run. After an initial relaxation time of about 200 fs, during which the initially localized polaron expands to its average size, the IPR, Eq. (5), was block-averaged over the remainder of the trajectories (Fig. 4b).

**Calculation of charge mobility**. While for the above samples of pMSB, PER, NAP, and ANT the 2D charge mobility tensor within the herringbone layer is converged, for the high-mobility OSs: RUB, DATT, and PEN, even larger system sizes would be required, which is currently still impractical. Importantly, we found that for the former set of systems the charge mobility along the $a$ ($b$) directions as obtained from the 2D mobility tensor are very well approximated by the charge mobility of a 1D chain of electronically active molecules along the $a$ ($b$) direction. We expect this correspondence to be even better for RUB, DATT, and PEN because the electronic coupling anisotropy and hence the preference for conduction in a single direction within the herringbone layer is more pronounced than e.g., for NAP and PER. To ensure consistent comparison between the simulated OSs we show in Fig. 4 the mobilities obtained for 1D chains of electronically active molecules in the indicated direction. The following number of molecules were used: 13 for pMSB; 25 for PYR; 20 for PER; 20 for NAP; 36 for ANT; 30 for RUB; 88 for DATT; 76 for PEN. We find that the mobilities are well converged for these system sizes, see Supplementary Fig. 7. For each system 1000 FOB-SH trajectories of length 1 ps were run. The mean-square displacement (MSD) of the charge carrier wavefunction was calculated according to Eq. (6),

$$\mathrm{MSD}(t) = \frac{1}{N_{traj}} \sum_{n=1}^{N_{traj}} \langle \Psi_n(t) | (x - x_0)^2 | \Psi_n(t) \rangle$$
$$\approx \frac{1}{N_{traj}} \sum_{n=1}^{N_{traj}} \left( \sum_{\nu=1}^{N_{mol}} |u_{\nu,n}(t)|^2 (x_{\nu,n}(t))^2 \right), \tag{6}$$

where $x$ is the position coordinate and $x_{\nu,n}(t)$ the time-dependent position of the center of mass of molecule $\nu$ in trajectory $n$ along the chain ($a$, $b$ or $c^*$ direction), and $x_0 = \langle \Psi_n(0)|x|\Psi_n(0)\rangle \approx x_{\nu=i,n}(0) = 0$. The MSDs averaged over FOB-SH trajectories are shown in Supplementary Fig. 3. After an initial relaxation time of a few 200 fs (as observed for the IPR), the MSD increases linearly indicative of Einstein diffusion. The Einstein diffusion coefficient is obtained from a linear fit of the data from about 0.5 to 1 ps,

$$D = \frac{1}{2} \lim_{t\to\infty} \frac{d\mathrm{MSD}(t)}{dt}, \tag{7}$$

and inserted in the Einstein relation for charge mobility,

$$\mu_{SH} = \frac{eD}{k_B T}, \tag{8}$$

where $e$ is the elementary charge, $k_B$ the Boltzmann constant and $T$ the temperature (300 K). The mobilities for these systems are well converged with respect to the chain length, as shown in Supplementary Fig. 7. We note that different definitions of MSD have been used in the literature for the calculation of charge mobility from explicit wavefunction propagation. In our previous work the displacement of the center of charge (coc) of the wavefunction was used,

$$\mathrm{MSD}_{coc}(t) = \frac{1}{N_{traj}} \sum_{n=1}^{N_{traj}} \langle \Psi_n(t) | (x - x_0) | \Psi_n(t) \rangle^2, \tag{9}$$

(simply denoted "MSD" in ref. [23]), whereas in refs. [16,17] the spread of the wavefunction ($\sigma$) was used,

$$\mathrm{MSD}_\sigma(t) = \frac{1}{N_{traj}} \sum_{n=1}^{N_{traj}} \langle \Psi_n(t) | (x - \langle x \rangle)^2 | \Psi_n(t) \rangle \tag{10}$$

with $\langle x \rangle(t) = \langle \Psi_n(t)|x|\Psi_n(t)\rangle$. We prefer the definition in Eq. (6) because it accounts for both types of diffusion, center of charge motion and spreading of the charge distribution:

$$\mathrm{MSD}(t) = \mathrm{MSD}_{coc}(t) + \mathrm{MSD}_\sigma(t). \tag{11}$$

Hence, the mobilities can be interpreted in terms of these two contributions as well,

$$\mu_{SH} = \mu_{SH,coc} + \mu_{SH,\sigma}. \tag{12}$$

For all systems investigated we find that $\mu_{SH} \approx \mu_{SH,coc}$ to a very good approximation since the average size of the polaron remains virtually unchanged after initial relaxation ($\mathrm{MSD}_\sigma \approx \mathrm{const}$). A detailed comparison between the different definitions will be reported in a forthcoming publication. The FOB-SH charge mobilities with all electronic couplings frozen to their mean coupling in a given direction,

$H_{kl}(t) = V = $ const, were obtained similarly (see Table 1 for values of $V$) and the results are shown in Fig. 4c.

**Dependence of mobility and IPR on the initial state**. As mentioned before, the system was equilibrated with molecule $i$ in the charged state and all other molecules in the neutral state and the charge carrier wavefunction was initialized accordingly as the frontier orbital localized on molecule $i$, $\Psi(0) = \phi_i(0)$. We also investigated other initializations of the system, e.g., starting from configurations where all molecules were initially equilibrated in the charge neutral state, and the charge carrier wavefunction is initially localized on a randomly chosen single molecule $m$, $\Psi(0) = \phi_m(0)$. While, obviously, the short-time dynamics differs in each case, after about 200 fs all of the differently initialized systems relax to the same polaronic state with the same IPR and mobility. In addition, we notice that detailed balance in FOB-SH algorithm (see Supplementary Fig. 1) ensures that even when the electronic carrier wavefunction is initialized as a superposition of localized states (for example an eigenstate of the Hamiltonian, namely $\Psi(0) = \psi_n(0)$), after the aforementioned relaxation period the charge carrier forms the same polaronic state and exhibits the same dynamics as in the case of the initially localized charge. This is an important observation because it shows that for calculation of charge mobility it is not necessary to reproduce the (usually unknown) initial state in experiment.

**Dependence of mobility on electric field**. In the present work the mobilities are calculated for the limit of zero external electric field. We have previously investigated the effect of electric field on carrier mobility in a simple 1D chain of chemically identical molecules[6]. We found that for a typical set of parameters for OSs, 50 meV coupling and 150 meV reorganization energy, the mobility remains independent on the applied external field up to field strengths of about $10^6$ V cm$^{-1}$. This is at least an order of magnitude higher than typical field strengths in time-of-flight (TOF) measurements ($10^3$–$10^5$ V cm$^{-1}$)[46]. Non-linear transport behavior may occur at higher field strengths, in which case the (field-dependent) diffusion coefficient in Eq. (8) could be obtained from the drift velocity–drift velocity autocorrelation function or from the derivative of the drift velocity with respect to the electric field strength[6].

**Detailed balance and internal consistency**. Detailed balance and internal consistency are two highly desirable attributes of any surface hopping (SH) simulations[22]. Detailed balance is achieved when the population of a given adiabatic electronic state $i$, $P_i^{\text{surf}}$, i.e., the fraction of time the nuclear dynamics runs on adiabatic potential energy surface $E_i$,

$$P_i^{\text{surf}} = \frac{1}{N_{\text{traj}}} \sum_{n=1}^{N_{\text{traj}}} \frac{1}{T_n} \int_0^{T_n} dt \, \delta_{ia,n}(t), \quad (13)$$

is equal to the Boltzmann population of this state,

$$P_i^{\text{B}} = \frac{\exp[-\Delta A_i/(k_\text{B}T)]}{\sum_j \exp[-\Delta A_j/(k_\text{B}T)]}. \quad (14)$$

In Eq. (13), $\delta_{ia,n}(t) = 1$ if state $i$ is the active surface $a$ on which the nuclear dynamics is running at time $t$ and equal to zero otherwise, and $T_n$ is the length of a trajectory. In Eq. (14), $\Delta A_i$ is the free energy difference between electronic state $i$ and the electronic ground state $i = 0$, which can be written as

$$\Delta A_i = -k_\text{B}T \ln\langle \exp[-\Delta E_i/(k_\text{B}T)]\rangle_{E_0}, \quad (15)$$

where $\Delta E_i$ is the vertical energy gap, $\Delta E_i(\mathbf{R}) = E_i(\mathbf{R}) - E_0(\mathbf{R})$. For each of the 1D chains described above we run 1000 independent equilibrium MD trajectories of length 1 ps in the electronic ground state $E_0$ to sample the exponential average, Eq. (15), and compared the resultant Boltzmann population, Eq. (14), to the surface populations obtained from FOB-SH, Eq. (13). The results for the first 9 adiabatic electronic states are shown in Supplementary Fig. 1. We find that the surface populations are in excellent agreement with the Boltzmann populations obtained from equilibrium MD simulation. As we have shown for smaller model systems before, the rescaling of the velocity component parallel to the NACV after a successful surface hop is essential to obtain good detailed balance[22]. The population of excited states within the band that is formed by the frontier orbitals of our molecules is significant for all systems suggesting that thermal excitations of the charge carrier cannot be neglected.

Internal consistency is achieved when the average quantum amplitudes of the electronic wavefunction $\Psi(t)$,

$$P_i^{\text{wf}} = \frac{1}{N_{\text{traj}}} \sum_{n=1}^{N_{\text{traj}}} \frac{1}{T_n} \int_0^{T_n} dt \, |c_{i,n}(t)|^2, \quad (16)$$

are equal to the surface population $P_i^{\text{surf}}$, Eq. (13). In Eq. (16), $c_i$ are the expansion coefficients of $\Psi(t)$ in the adiabatic (i.e., electronic eigenstate) basis, $\Psi(t) = \sum_{i=1}^{M} c_i(t)\psi_i(\mathbf{R}(t))$, where $c_i(t)$ are related to $u_i(t)$ of Eq. (1) by the unitary transformation that diagonalizes the electronic Hamiltonian. We find excellent agreement between the two populations for the lowest five electronic states

(down to populations of $10^{-2}$), with some minor deviations for higher lying states, see Supplementary Fig. 1. The latter probably arises from the projection algorithm that we use to remove decoherence correction-induced artificial long-range charge transfer events. Yet, since the small deviations only occur for states with low population this small discrepancy should have no significant effect on our results.

**Importance of decoherence correction**. In Supplementary Fig. 2 we show results for hole transport in anthracene when the decoherence correction is switched off. First we note that there is no longer internal consistency, quite the opposite: the quantum population is almost the same for all electronic states, i.e., the temperature of the electronic subsystem becomes infinite, the infamous problem of the original Ehrenfest and SH methods[47]. A consequence of this is that the polaron size (IPR) and the mobility are strongly overestimated because most of the higher lying electronic states that are now occupied are more delocalized than the lower lying states. Even more seriously, the IPR and charge mobility do not converge with system size. For all these reasons it is of utmost importance to apply decoherence correction, otherwise the energy level population and the charge transport dynamics becomes unphysical.

**Charge mobility from small polaron hopping**. Charge mobilities were calculated for a small polaron model for hopping between nearest neighbors (M) within the 1D chain (green data points in Fig. 4a), e.g., for hole transfer,

$$\text{M}_i^+ - \text{M}_{i+1} - \text{M}_{i+2} - \underset{k_{ii+1}}{\overset{k_{i+1i}}{\rightleftharpoons}} \text{M}_i - \text{M}_{i+1}^+ - \text{M}_{i+2} - \underset{k_{i+1i+2}}{\overset{k_{i+2i+1}}{\rightleftharpoons}} \cdots$$

where $k$ is the rate constants obtained from electron transfer theory assuming equivalent sites. We adopted the following semiclassical transition state theory formula valid in the non-adiabatic and adiabatic ET regime[6]

$$k = \kappa_{\text{el}} \nu_{\text{eff}} \exp(-\beta \Delta A^{\ddagger}), \quad (17)$$

where $\beta = 1/k_\text{B}T$, $\kappa_{\text{el}}$ is the electronic transmission coefficient,

$$\kappa_{\text{el}} = \begin{cases} \frac{2P_{\text{LZ}}}{1+P_{\text{LZ}}} & \text{if} \quad \Delta A \geq -\lambda \\ 2P_{\text{LZ}}(1-P_{\text{LZ}}) & \text{if} \quad \Delta A < -\lambda \end{cases} \quad (18)$$

$$P_{\text{LZ}} = 1 - \exp(-2\pi\gamma) \quad (19)$$

$$2\pi\gamma = \frac{\pi^{3/2}\langle |H_{kl}|^2\rangle}{h\nu_{\text{eff}}\sqrt{\lambda k_\text{B}T}}, \quad (20)$$

$\nu_{\text{eff}}$ is the effective nuclear frequency (in our system, taken to be the stretching frequency of an aromatic carbon double bond: 1600 cm$^{-1}$) and $\Delta A^{\ddagger}$ is the activation barrier. For vanishing driving force, $\Delta A = 0$, as is the case here, $\Delta A^{\ddagger}$ is given by[32],

$$\Delta A^{\ddagger}(\Delta A = 0) = \frac{\lambda}{4} - \left(V - \frac{V^2}{\lambda}\right). \quad (21)$$

$\Delta A^{\ddagger}$ and $\kappa_{\text{el}}$ were evaluated for the same reorganization energy $\lambda$ used and mean couplings $V$ obtained from FOB-SH simulations (Table 1). The time evolution of the population for each site can be found solving the first order differential equation:

$$\frac{d\mathbf{P}(t)}{dt} = \mathbb{K}\mathbf{P}(t) \quad (22)$$

where $\mathbf{P}(t)$ is a vector containing site populations and $\mathbb{K}$ is the matrix of rate constants. The latter takes the general form:

$$\mathbb{K} = \begin{bmatrix} -k & k & 0 & 0 & \ldots & 0 \\ k & -2k & k & 0 & \ldots & 0 \\ 0 & k & -2k & k & \ldots & 0 \\ 0 & 0 & k & -2k & \ldots & 0 \\ \vdots & \vdots & \vdots & \vdots & \ddots & \vdots \\ 0 & 0 & 0 & \ldots & k & -k \end{bmatrix}$$

The solution to Eq. (22) is

$$\mathbf{P}(t) = \exp(\mathbb{K}t)\mathbf{P}(0) \quad (23)$$

where $\mathbf{P}(0)$ is the vector of initial populations, in our case the first component $P_1(0) = 1$ and all other components are zero. The MSD is then obtained through

$$\text{MSD}(t) = \sum_\nu P_\nu(t)(\nu L)^2, \quad (24)$$

where $L$ is the distance between the center of mass of two neighboring molecules and $\nu$ the index of the molecule.

**Table 3 Computed and experimental mobilities (in $cm^2V^{-1}s^{-1}$)**

| crystal | dir. | dist. | $\mu_{SH}$[a] | $\mu_{SH}$ Frz.V[b] | $\mu^*_{TLT}$[c] | $\mu_{TLT}$[d] | $\mu_{hop}$[e] | $\mu_{band}$[f] | $\mu_{exp}$[g] |
|---|---|---|---|---|---|---|---|---|---|
| DATT-h$^+$ | $a$ | 6.26 | 10 ± 1.1 | 35 ± 7.3 | 5.5 | 6.3 | – | 322.6[h] | 16[i] |
| RUB-h$^+$ | $a$ | 7.18 | 4.9 ± 0.18 | 33 ± 3.3 | 1.4 | 3.8 | – | 51[j] | 9.7(8–20)[k] |
| PEN-h$^+$ | $T_1$ | 4.80 | 9.6 ± 1.8 | 26 ± 3.3 | 4.5 | 5.3 | – | 58[j] | 10.5(5–35)[l] |
| ANT-h$^+$ | $a$ | 5.24 | 0.86 ± 0.05 | – | 0.26 | 0.40 | 1.5 | 19.2[m] | 1.1[n,o] |
|  | $b$ | 6.04 | 1.9 ± 0.17 | 8.8 ± 0.50 | 0.77 | 1.2 | 5.0 | 42.2[m] | 2.9[n,o] |
| NAP-h$^+$ | $b$ | 5.95 | 1.3 ± 0.05 | 2.3 ± 0.39 | 2.4 | 3.7 | 1.8 | 74.4[m] | 1.3[p] |
| PER-e$^-$ | $a$ | 6.10 | 2.4 ± 0.09 | 3.4 ± 0.13 | 1.3 | 2.3 | 2.8 | – | 2.3[q] |
|  | $c^*$ | 10.26 | 0.31 ± 0.08 | 0.26 ± 0.04 | – | – | 0.37 | – | 0.27[r] |
| PYR-e$^-$ | $c^*$ | 8.47 | 0.62 ± 0.12 | 0.44 ± 0.03 | – | – | 0.53 | – | 0.51[s,o] |
| pMSB-h$^+$ | $b$ | 5.88 | 0.21 ± 0.03 | 0.16 ± 0.02 | 0.35 | 2.4 | 0.16 | – | 0.17[t] |

[a]FOB-SH mobility
[b]FOB-SH mobility, with electronic couplings frozen to their thermal average $V$ (Table 1)
[c]Mobility from TLT without modification of the onsite energies
[d]Mobility from TLT after setting all onsite energies to zero at all times
[e]Small polaron hopping mobility using the semiclassical ET rate (see Methods)
[f]Band mobility taken from the literature
[g]Experimental mobilities. Where available, time of flight (TOF) experimental measurements in bulk crystals in the specified crystallographic direction have been used in order to match as closely as possible our simulation conditions
[h]Ref. [48]
[i]Ref. [49] Field-effect-transistor (FET) mobility, surface of $P2_1$ single crystal, the reported structure corresponds to CSD entry: AVIBEN
[j]Ref. [50]
[k]Ref. [51] Hall-effect mobility, surface of single crystal grown by physical vapor transport in hydrogen. (According to Ref. [52] Hall-effect measurements are highly desirable when relatively high mobility organic semiconductors are investigated). In parenthesis is given the range of experimental mobilities reported in the literature[53–57]. In ref. [55] orthorhombic polymorph with lattice constant along $a$ and $b$ directions of 7.2 and 14.4 Å, respectively, is examined
[l]Ref. [58] space-charge-limited current (SCLC) mobility as measured in bulk. In parenthesis is given the experimental range in which FET mobilities have been measured on the OS surface according to refs. [59,60]. In ref. [59] polymorph with thickness d(001) equal to 14.1 Å is examined
[m]Ref. [61]
[n]Ref. [46] TOF measurement, bulk of the crystal
[o]Usually crystallizes as the $P2_1/a$ polymorph.
[p]Refs. [62,63] TOF measurement, bulk of the $P2_1/a$ polymorph crystal
[q]Ref. [64] TOF measurement, bulk of the $\alpha$-perylene polymorph crystal
[r]Ref. [65] TOF measurement, bulk of the crystal
[s]Ref. [66] TOF measurement, bulk of the crystal
[t]Refs. [67,68] FET mobility, surface of the crystal, the reported structure corresponds to CSD entry: LUJSAL

**Charge mobility from transient localization theory**. We have calculated $\mu_{TLT}$ and $\mu^*_{TLT}$ along the directions specified in Table 3 using the electronic Hamiltonian sampled in FOB-SH trajectories. For the calculation of $\mu_{TLT}$ onsite energies are set to zero as done in refs. [14,15], whereas for the calculation of $\mu^*_{TLT}$ the complete Hamiltonian including onsite thermal fluctuations is used. We employed the exact diagonalization method proposed in ref. [15] to calculate the squared transient localization length along $x$ and $y$ direction in the 2D herringbone layer of the investigated systems, $L^2_{x(y)}$, Eqs. (8) and (9) in ref. [15]. The intermolecular oscillation time $\tau = 1/\omega_0$ is taken as the inverse of the angular frequency $\omega_0$ of the highest peak in the power spectrum of the electronic coupling time series evaluated along 5 ps long FOB-SH trajectories (summarized in Table 2). Mobilities from TLT are shown in Fig. 5a. The average squared localization length in the 2D plane was divided by the area $A$ per molecule within the herringbone layer to enable comparison with the IPR obtained from FOB-SH, $L^2_\tau/A = (L^2_x + L^2_y)/2A$ (see Fig. 5b).

## Data availability

All data supporting the findings of this study are available from the corresponding author upon request. All the custom codes used in this study are available from the corresponding author under reasonable request.

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

## Acknowledgements

We thank Dr. Guido Falk von Rudorff for helpful discussions. S.G., A.C., O.G.Z., and S.Ghosh were supported by the European Research Council (ERC) under the European Union, Horizon 2020 research and innovation programme (grant agreement no. 682539/ SOFTCHARGE). Via our membership of the UK's HEC Materials Chemistry Con-sortium, which is funded by EPSRC (EP/L000202, EP/R029431), this work used the ARCHER UK National Supercomputing Service (http://www.archer.ac.uk), as well as the UK Materials and Molecular Modeling (MMM) Hub, which is partially funded by EPSRC (EP/P020194), for computational resources. We also acknowledge the use of the UCL Grace High Performance Computing Facility.

## Author contributions

S.G., A.C., M.E., O.G.Z., S. Ghosh enhanced FOB-SH code capabilities and accuracy. M.E. developed wavefunction visualization code. S.G., H.Y., and O.G.Z. prepared the systems and parametrized force-fields. S.G. and O.G.Z. performed DFT calculations and set-up FOB-SH parameters. S.G. and A.C. performed actual simulations and wrote post-processing tools. J.B. designed the research and helped with data interpretation. S.G. and J.B. wrote the manuscript. All authors reviewed and discussed the manuscript.

## Additional information

**Competing interests:** The authors declare no competing interests.

