## [Peer Review File · Nature Communications]

Reviewers' comments:

Reviewer #1 (Remarks to the Author):

Publish with some revisions. Please see attached review pdf for details

Reviewer #2 (Remarks to the Author):

Review of "Quantum Localization and Delocalization of Charge Carriers in Organic Semiconducting Crystals" by Samuele Giannini et al for Nature Communications.

The manuscript by Giannini et al. presents the use of a model based on solving the time-dependent electronic Schrodinger equation to simulate charge transport in a series of organic semiconductor crystals. The model is based on a previously published approach by the same group, the fragment-orbital based surface hopping (FOB-SH) method, but has been substantially enhanced in this work. The study reports, for each crystal studied, the evolution in time and space of the wavefunction of a polaron state, and discusses how the spatial extent and dynamics of the polaron in the different systems relate to the simulated hole diffusion coefficient. Moreover the simulated data are compared with experimentally reported mobilities on the same set of compounds and agree very well. The observed behaviour is qualitatively similar to that described by the 'transient localisation' approach of Fratini and others in that charges move via a series of jumps from one – distributed – location in space to another, but this work goes far beyond any I have seen before in that it illustrates how differences in transport behaviour can be related to the chemical structure and spatial interactions of the molecules, and illustrating the time dependent evolution of the wavefunction on timescales shorter than the typical hop. The paper is also very clearly written and presented. The results are very well supported by a detailed methods section which will allow other readers to follow and use the approach, which will be appreciated by the community.

I find the paper to be excellent and absolutely deserving of publication in Nature Communications, subject to attention to some minor issues listed below. In my opinion it is significant enough to merit publication in higher impact journals in the Nature group.

The authors should consider the following points in a revised version:

1. This information may be given somewhere but I could not find it: how well do the authors know that the crystal structures they used for each of the eight semiconductors were the dominant polymorphs for the samples used in the experiments from which the experimental TOF or FET mobility measurements were taken? I expect the answer is known to a different degree of certainty in the different cases, but it would be helpful to know in which cases (or if in all cases) the crystal structure of the measured sample is reported.

2. Although the paper is about charge carrier mobility (response to an electric field) the concept of electric field seems not to be mentioned anywhere in the paper. It would be interesting to have some comment on how the presence of an electric field could affect the results (would non-linear behaviour be expected?) and how it might be incorporated in future developments of the model. Also the validity of the Einstein diffusion approximation for these materials should be commented on in the text.

3. On p. 4 line 64, please specify that the reorganisation energy calculated is the inner one. In the Supporting Information the authors argue that the outer contribution to reorganisation energy is negligible. Would that remain valid in the case of disordered crystals or molecules of more polar character?

4. Could the authors comment on how the choice of the basis set and functional in the DFT calculation affects the results of the model. From my understanding the authors extract parameters of the model such as the force field and the geometry of the molecules using DFT, could they explain how robust the results are to the choice of theoretical method?

5. In the current contribution the authors mostly focus on modelling the system under room temperature. Could the authors comment on the validity of the approach for different temperatures, and, if possible, comment on the degree to which the model reproduces the observed temperature dependence? Since the relationship between temperature dependence of mobility and transport mode in organic semiconductors has been a widely visited topic in literature, the predictions of this model for temperature dependence could improve the appeal of the paper.

6. From the model presented, the authors only model the evolution of a single excited state. Can the model be applied (or adapted for) studies of the dynamics of more than one excited state in the same supercell?

Minor remarks:

7. "CT" is very commonly used to refer to charge transfer in molecular semiconductors rather than charge transport. Is it possible to avoid needing to use the acronym "CT"?

8. Figure 4b, although interesting, is not referred to in the main text.

Reviewer #3 (Remarks to the Author):

The manuscript by Giannini and collaborators tackles an important open problem in the field of organic electronics: understanding the underlying principles of hole conduction in molecular organic semiconductors (OS). This problem has been a true head-scratcher for the community. There are a plethora of possible theories that can be applied for modeling hole conductivity. However, each one of them has shown fallacies in predicting one or multiple experimental observations. For the past 5-10 years it has been clear that OS sample a very rich grey area of dynamics mechanisms. Thus, clarity is dearly needed.

This manuscript delivers the needed clarifications. Using known descriptors (e.g., λ and V), the authors describe in minute detail the interesting and multifaceted dynamics that occurs in the initial steps of hole transfer when the initial state is a hole localized on a simple molecule. The authors describe a dynamics that is reminiscent of certain types of biological charge transfer even though there are some clear distinguishing features compared to bio systems (such as 3-dimensionality). I am convinced that this is a quality manuscript that has the chance of being cited hundreds of times, and to become the next "talk of the town".

In doing my due diligence, I find several aspects, especially of the electronic structure/dynamics methodology that can be explained more in depth:

1. In the interest of a more self-contained publication, the authors should describe their method for generating the position-dependent electronic Hamiltonian matrix. And they should clearly point out any possible shortcoming of the method. This will help other researchers determine what can be improved in the method and whether the predictions can be further crystallized by employing more accurate electronic structure methods.

2. The authors should mention that they have not carried out a study of the initial state dependence. There are conjectures that the initial state of a model should somehow mimic the one

realized by experiments. Clearly that is another daunting task. But I wonder if the authors can shed some light on this, or perhaps simply acknowledge this.

3. Figure 2 can be improved. I understand that the authors are trying to convey the most information with the least space. I find the figure hard to interpret and perhaps unnecessarily complex.

4. A typical method to include modulation of the Hamiltonian matrix elements with nuclear motion is to model them by an average value and a random distribution of variations from the average with a width that is typically either guessed, or evaluated by MD. If such an approach is employed with appropriate parameters extracted from the author's simulations (say, from Table S1), would it recover the same dynamics and conclusions? Can the authors make a comment along these lines in the manuscript?

Reviewer 1:

The submitted manuscript is an important contribution to the study of transport in organic semiconductor crystals. The work uses the recently developed DFT-based FOB-SH method of mixed quantum-classical nonadiabatic dynamics which enables the ab-initio simulation of transport in large systems for sufficiently long time scales so as to explore very different transport regimes. In this study the FOB-SH simulations are applied to a variety of semiconductor crystals that exhibit very different mobilities and are likely to involve very different transport mechanisms. The simulations reproduce well the experimental mobilities of these systems, demonstrating the power of the computational method. Importantly, the study attempts to connect the mobilities to relevant electronic-structure parameters of the systems and to interpret and connect the simulation results to simple reduced models of transport for the different regimes. The paper should certainly be published as it is a significant contribution to the field, but there is a need for some improvements in the presentation which will make the results more transparent to a general audience and more useful to theorists pursuing reduced models of transport.

The authors convincingly correlate increase in mobility values to the increase in two parameters, the average inverse participation ratio (IPR) and the ratio $\xi=2V/\lambda$ of the nearest neighbor electronic coupling to the reorganization energy. Their conclusion is that high IPR and ξ (i.e. highly delocalized charge-carrier states) maximize the mobility. In the highest mobility regime they show that the recent transient localization theory reproduces their simulation results and in the lowest mobility regime a nearest-neighbor hopping model is adequate, although it is possible that the flickering resonance mechanism also becomes relevant. The weakest part of the presentation is in the discussion of the low to medium mobility regime and the possible underlying mechanisms beyond nearest-neighbor hopping (i.e. flickering resonance related mechanisms). This part of the presentation should be refined.

1) The presentation and analysis would benefit from the inclusion of more statistical results, some of which involving time-scale parameters. After all, an advantage of the FOB-SH method is that it can be used as a tool for extracting transport statistics both in the energy and time domains. Figures 2 c and k show single trajectory IPR fluctuations and average IPR values as a function of time for two of the materials. In both

figures the single trajectory IPR plots show low IPR and high IPR regions with comparable time periods. Are these observations general? The author should provide in the main text:

1 a) standard deviations of IPR values for the systems.

1 b) average lifetimes of high IPR fluctuation events (high compared to average), b) average time scales between high IPR fluctuation events. This information will provide insight about the role of flickering resonance activation events in the low to medium mobility regimes.

1 c point C) None of the above discussion is relevant to the flickering resonance mechanism where the rate limiting factor may be due to the activation step for flickering resonance conformations which involve ballistic transport over multiple sites. This fact should also be pointed out and the time-domain statistics requested in comment (1) about activation events should provide some info about the flickering resonance mechanism.

We address comments 1a, 1b and 1c point C together. We have included a new table (Table 2 in revised version) summarizing the mean and root-mean-square fluctuations of IPR, as well as the average duration of and time between resonances. These additional results are mentioned/discussed in the main text: we mention the average and root-mean square fluctuations on page 4 for pMSB “The average IPR is equal to 1.7 and the root-mean-square fluctuation σ equals 0.9 (see Table 2) and for pentacene “On average, the polaron is delocalized over 18 molecules ($\sigma = 10.3$) in excellent agreement with estimates based on experimental electron spin resonance data, 17 molecules at 290 K”. We have also introduced a new paragraph on page 5 and a new Figure 3 analysing in detail the statistics and time scales of the resonance events: “The dynamics at longer times, up to a few picoseconds, is shown in Figure 3 for both materials. We find that the average duration of a “resonance”, defined here by the time it takes for the IPR to exceed and subsequently return below $\langle \mathit{IPR} \rangle + \sigma$ is 7 and 12 fs for pMSB and pentacene, respectively, see Figs. 3a and 3b, which is close to the characteristic oscillation time of intramolecular vibrations and site energy fluctuations. The average time between two resonances is about an order of magnitude larger, 52 fs for pMSB and 114 fs for pentacene. Similar values are obtained for the other compounds, see Table 2. These resonances give rise to spatial displacements as described qualitatively above and shown in Fig. 3 by way of projecting $\Psi(t)$ on the high-mobility crystallographic directions b and $T1$ of pMSB (Fig. 3c) and pentacene (Fig. 3d), respectively. Yet, significant displacements along the high mobility directions occur at somewhat longer times than the time between two resonances, more characteristic of the oscillation time of the electronic coupling fluctuations, $\tau = 1159$ and 202 fs/rad for pMSB and pentacene, respectively, see Figs 3c and 3d. Hence, as one would expect, only a fraction of the resonances (estimated to be about 0.2-0.5) leads to a successful displacement. Notably, the wavefunction displacements in pentacene are over several lattice spacings at a time, 3-5 nm, that is about an order of magnitude larger than the (mostly nearest-neighbour) displacements in pMSB. As we will see in the following, this difference gives rise to a ≈ 50 -fold higher charge mobility in pentacene relative to pMSB. “ The actual dynamics in the hopping regime is more complicated than any of the idealised model mechanisms including flickering resonance. For instance, when the carrier becomes temporarily delocalized, the charge may relocalize on any of the sites that are in resonance, not necessarily the one furthest apart. Moreover, in the present dynamics, charge delocalization and displacements are often associated with electronic transitions (“surface hops”) of the nuclei onto higher-lying adiabatic eigenstates states (as pointed out on page 5). This makes the interplay between energy resonances and localization/delocalization/displacements of the carrier even more intricate. For all these reasons, we prefer to say that in the hopping regime there is a qualitatively reminiscence with flickering resonance, as mentioned in the manuscript, but prefer not to draw the parallel much further, at least in the current work. We agree with the referee that it might be possible the develop refined reduced models based on the FOBSH data presented herein, and we leave this for future investigations.

1 c) The modelling and discussion of small to medium polaron transport via a hopping mechanism needs improvement because it is confusing to the non-expert. The authors suggest that the medium polaron hopping regime is fortuitously reproduced by a nonadiabatic hopping model. The use of the word fortuitous seems to be due the fact that a small-polaron nonadiabatic hopping model reproduces medium-polaron transport.

A) The discussion of the hopping modelling for the fortuitous regime is quite vague for the nonexpert.

More details should be added (also in the SI) rather than referring to previous papers.

B) It is known (e.g. *Nature Nanotechnology* 9, 1040–1046 (2014) that a small polaron hopping model may give the same order of magnitude current as a larger polaron model: in a 1-D molecular chain of a fixed length (such as DNA), reducing the number of effective hopping sites (going from a highly localized to more delocalized charge carrier states) will retain the same overall current for a fixed distance if simultaneously the effective-site to effective-site hopping rates are lowered. From a physical point of view, in a 1 D system, delocalizing (diluting) the effective hopping site from 1 to N units will approximately reduce the effective-site to site electronic coupling by N and the reorganization energy by N and thus affect the overall nonadiabatic rate. Eventually the effect of delocalization will be to switch the hopping rate from nonadiabatic to fully adiabatic for which the effective-site to site coupling is irrelevant (not rate limiting). Thus, different hopping models may give the same overall currents and mobilities for a given length and hopping model fitting by itself does not provide sufficient info about the nature of the hopping-rate mechanism. These finer points which the authors are aware of, should be discussed a bit more for the non-expert rather than using the word fortuitous.

We address 1c point A and B together. We have revised and expanded our discussion of the small-to-medium polaron hopping regime as recommended and included the reference to the *Nature Nanotechnology* 2014 paper. We have introduced a new paragraph at the end of page 6: “For OSs with larger mobilities, $\approx 1-5 \text{ cm}^2 \text{ V}^{-1} \text{ s}^{-1}$ ($0.2 < \xi < 1$), the free energy barrier is small, causing the polaron to delocalize over 2-5 molecules according to FOB-SH simulations. Hence, in this regime the small polaron hopping model assuming nearest neighbour hops of a fully localized charge carrier is no longer a good physical model of the charge transport process. Nonetheless, if one solves the chemical Master equation with nearest neighbour hopping rates from ET theory, the resultant mobilities are in good agreement with FOB-SH and experimental values (data in shaded green). This agreement appears to be coincidental as the small polaron hopping mechanism bears no resemblance with the actual mechanism obtained from FOB-SH. Indeed, it is well known that a small polaron hopping model may give the same order of magnitude in mobility or current as a larger polaron model¹⁴ - agreement with the experimental mobility gives no sufficient information on the mechanism.” As requested, we have added more detail in the methods section regarding the calculation of the ET rate Eq. 17 (which is valid in the non-adiabatic and adiabatic regimes). We added Eq. 18-20 and 22-24.

4) A central assumption of the transient localization theory is the relaxation time approximation, i.e., the assumption of exponential decay wrt time of the retarded current-current anticommutator correlation function. It would be very interesting and important to show the time evolution of this correlation function for the high-mobility materials (if this is currently feasible computationally).

We agree with the reviewer that this might be an interesting point, however unfortunately we do not have access to the electronic velocities in the current version of our FOB-SH algorithm, as we are not calculating the electronic gradient of the carrier wavefunction. While our intension was to learn how well transient localization theory (TLT) can capture the results of current numerical simulations, we feel that a full investigation of the assumptions of TLT is beyond the scope of this work.

Reviewer 2:

The manuscript by Giannini et al. presents the use of a model based on solving the time-dependent electronic Schrodinger equation to simulate charge transport in a series of organic semiconductor crystals. The model is based on a previously published approach by the same group, the fragment-orbital based surface hopping (FOB-SH) method, but has been substantially enhanced in this work. The study reports, for each crystal studied, the evolution in time and space of the wavefunction of a polaron state, and discusses how the spatial extent and dynamics of the polaron in the different systems relate to the simulated hole diffusion coefficient. Moreover the simulated data are compared with experimentally reported mobilities on the same set of compounds and agree very well. The observed behaviour is qualitatively similar to that described by the ‘transient localisation’ approach of Fratini and others in that charges move via a series of jumps from one – distributed – location in space to another, but this work goes far beyond any I have seen before in that it illustrates how differences in transport behaviour can be related to the chemical structure and spatial interactions of the molecules, and illustrating the time dependent evolution of the wavefunction on timescales

shorter than the typical hop. The paper is also very clearly written and presented. The results are very well supported by a detailed methods section which will allow other readers to follow and use the approach, which will be appreciated by the community.

I find the paper to be excellent and absolutely deserving of publication in Nature Communications, subject to attention to some minor issues listed below. In my opinion it is significant enough to merit publication in higher impact journals in the Nature group.

The authors should consider the following points in a revised version:

1) This information may be given somewhere but I could not find it: how well do the authors know that the crystal structures they used for each of the eight semiconductors were the dominant polymorphs for the samples used in the experiments from which the experimental TOF or FET mobility measurements were taken? I expect the answer is known to a different degree of certainty in the different cases, but it would be helpful to know in which cases (or if in all cases) the crystal structure of the measured sample is reported.

We have checked the papers where the experimental mobilities are reported and only in two cases (DATT and pMSB) the exact crystal structure was explicitly stated. We added this information in the **footnotes of Table 3 in the revised version as well as additional structural information on the other compounds where available**. In particular, at ambient conditions, NAPH, ANT and PYR are known to crystallize into their stable monoclinic $P2_1/a$. In the case of NAPH mobility, in Ref 79 and 81 (main text), Warta et al. explicitly refer to $P2_1/a$ polymorphs grown by Bridgman techniques, although the exact crystal structure is not given. Concerning PERY, mobility is measured for α -polymorph as explicitly stated in Ref 81. In fact, under conventional conditions perylene is known to crystallize into this polymorph (Yago T. et al. *Chemistry Letters* **36**, 370-371 (2007)). In RUB and PEN the degree of uncertainty on the crystal structures used is a little larger and so probably does the uncertainty in mobility measurements available (that is why we indicated those measurements with error bars in the manuscript's figures). Regarding Rubrene, Podzorov and co-workers (Ref 73) usually refer to lattice constant along a and b directions of 7.2 and 14.4 Å, respectively and the orthorhombic structure when they measure mobility. This is practically the same as the structure in Ref. 56 that we have used. Pentacene has 4 known polymorphs, classified by the molecular layer thickness $d(001)$: 14.1, 14.4, 15.0, and 15.4 Å. The polymorph with thickness $d(001)$ 14.1 is the commonly adopted structure in single crystal (Mattheus, et al. *Crystal. Section C: Crystal Structure Comm.* **57** 939-941, (2001)) and Jurchescu et al., in Ref. 77, explicitly refer to this structure when measuring mobility. Thus, we adopted this polymorph reported in Ref. 57.

2) Although the paper is about charge carrier mobility (response to an electric field) the concept of electric field seems not to be mentioned anywhere in the paper. It would be interesting to have some comment on how the presence of an electric field could affect the results (would non-linear behaviour be expected?) and how it might be incorporated in future developments of the model. Also the validity of the Einstein diffusion approximation for these materials should be commented on in the text.

With regard to the second point, the Einstein diffusion approximation is valid because all MSDs are linear in the long-time regime to a very good approximation, as shown in Fig. S3. We have made this point more explicit on page 6 “After a short initial relaxation period, we observe a linear increase of the MSD with time, **implying that the Einstein diffusion approximation is valid.**” The effect of electric field on carrier mobility has been investigated and discussed in our previous work: Oberhofer, H., Reuter, K. & Blumberger, *Chem. Rev.* **117**, 10319–10357 (2017). We have added a short paragraph on page 19: “**Dependence of mobility on electric field.** In the present work the mobilities are calculated for the limit of zero external electric field. We have previously investigated the effect of electric field on carrier mobility in a simple 1D chain of chemically identical molecules.\cite{Oberhofer17} We found that for a typical set of parameters for OSs, 50 meV coupling and 150 meV reorganization energy, the mobility remains independent on the applied external field up to field strengths of about 10^6 V/cm. This is at least an order of magnitude higher than typical field strengths in time-of-flight (TOF) measurements (10^3 - 10^5 V/cm)\cite{Karl01}. Non-linear transport behaviour may occur at higher field strengths, in which case the (field-dependent) diffusion coefficient in Eq. 8 could be obtained from the drift velocity-drift velocity autocorrelation function or from the derivative of the drift velocity with respect to the electric field strength\cite{Oberhofer17}.”

3) On p. 4 line 64, please specify that the reorganisation energy calculated is the inner one. In the Supporting Information the authors argue that the outer contribution to reorganisation energy is negligible. Would that remain valid in the case of disordered crystals or molecules of more polar character?

We have added this detail on page 4 “We note in passing that reorganization energy is assumed to be equal to the intramolecular (or “inner-sphere”) contribution. The intermolecular (or “outer-sphere”) contribution is typically very small in apolar OS\cite{Oberhofer12,McMahon10} studied here and is neglected.” According to the Marcus expression, the outer-sphere reorganization free energy increases with increasing static dielectric constant of the material. Hence, while we expect that outer-sphere reorganization energy can still be neglected in apolar/non-hydrogen bonded disordered materials, we expect this no longer to be a good approximation for molecules with more polar/hydrogen-bonded character. We added a sentence on page 12: “We expect that this is no longer a good approximation for crystals formed of polar or hydrogen bonded molecules. In this case the full electrostatics including electronic polarization of the molecules should be included as it is well known that site energy fluctuations and hence reorganization free energies are overestimated for fixed point-charge models.\cite{Seidel09,Oberhofer10acie,Moens10}”.

4) Could the authors comment on how the choice of the basis set and functional in the DFT calculation affects the results of the model. From my understanding the authors extract parameters of the model such as the force field and the geometry of the molecules using DFT, could they explain how robust the results are to the choice of theoretical method?

This is correct, we use parameters from DFT calculations (specifically, reorganization energies and electronic couplings) to construct the electronic Hamiltonian for FOB-SH simulation. We have added a brief discussion on the robustness of reorganization energy with respect to choice of density functional and basis set on page 11. “Hybrid functionals are known to give good equilibrium structures and better energies for bond stretching than GGA functionals, which is important for the calculation of reorganization energies.\cite{Malagoli04,Yavuz15,Yang17} Taking anthracene as example, we obtain similar values for two of the most popular hybrid functionals, λ = 142.1 meV for B3LYP and 149.9 meV for PBE0, but a smaller value for the GGA functional BLYP, 102.4 meV, due to the well known deficiency of the latter functional to underestimate the energy for bond stretching. The results are well converged with respect to the basis set used. Only very small changes in λ are obtained as the basis set is increased: 138.1, 142.1 and 138.5 meV for the 6-31G(d), 6-311G(d) and 6-311G+(d,p) basis sets, respectively, using the B3LYP functional.” Our FODFT method for electronic coupling calculation has been extensively benchmarked before using the same basis set and functional as used in current calculations. We have added more detail on page 12 “Using the same functional and basis set, the accuracy of FODFT couplings was benchmarked before on the HAB11\cite{Kubas14jcp} and HAB7- databases\cite{Kubas15pccp} for electronic coupling for hole and electron transfer in π -conjugated organic dimers. While the mean relative unsigned error with respect to high-level ab-initio reference values was found to be reasonably small (27.9%), the values were slightly but uniformly underestimated. Hence, as recommended in the previous studies, we scaled the FODFT couplings for hole and electron transfer systems by a factor of 1.348 and 1.325 to obtain best estimates.” We have also added in Table 1 FODFT couplings obtained for OS crystal pairs along reported directions (crystals structures used are referenced in the footnotes of the same table).

5) In the current contribution the authors mostly focus on modelling the system under room temperature. Could the authors comment on the validity of the approach for different temperatures, and, if possible, comment on the degree to which the model reproduces the observed temperature dependence? Since the relationship between temperature dependence of mobility and transport mode in organic semiconductors has been a widely visited topic in literature, the predictions of this model for temperature dependence could improve the appeal of the paper.

We agree with the reviewer that temperature dependence of mobility is a property of interest, even though for practical applications the room temperature mobility, investigated in current work, is arguably the most

important. Experimentally, the motivation to determine the temperature dependence is to get some clue about the transport mechanism: a negative slope would indicate a “band-like” and a positive slope an activated “hopping-like” mechanism. However, this interpretation is not always valid, e.g. it is well known that for certain combinations of parameters characterizing OS the hopping mechanism can also produce a negative slope, see e.g. *PCCP* 14, 13846 (2012). The advantage of the FOB-SH methodology is that the mechanism is directly observable from the simulation, without reference to its temperature dependence. When calculating the temperature dependence one needs to bear in mind that mobility is sensitive but not very sensitive to changes in temperature compared to the statistical error of mobility in FOB-SH simulations, see Fig S7. One would need to go to relatively low temperatures (100 K or lower) to determine with statistical significance the temperature dependence from FOB-SH simulation. One can expect that at such low temperatures the classical nuclear dynamics in FOB-SH may no longer be a good approximation. In preliminary FOB-SH calculations we found that the temperature dependence is in good agreement with experiment for high mobility OS (rubrene, pentacene). Yet, for low mobility OS (anthracene etc) nuclear quantum effects, in particular nuclear tunnelling, are important, which is not included in the FOB-SH methodology. A methodology that includes these effects is currently under development in our group (surface hopping with quantum nuclei) but not ready yet for application to condensed phase systems. We believe it best to publish the calculated temperature dependences in a separate study once this extension to the FOB-SH methodology can be applied to the systems studied here. A first application of the extension to charge transfer in a pi-conjugated model dimer in the gas phase indicated that nuclear quantum effects on the ET rate are negligible at room temperature but become important at low temperatures (Ghosh S., Giannini S., Blumberger J, *Faraday Discussion*, 2019, *Just accepted DOI: 10.1039/C9FD00046A*). This justifies our statement made on page 3 “... (FOB-SH) method\cite{Spencer16jcp,Carof17,Giannini18} is a truly predictive approach in this regard, in particular at ambient and high temperature where nuclear quantum effects are still relatively small.” **We now cite the above paper after this statement.**

6) *From the model presented, the authors only model the evolution of a single excited state. Can the model be applied (or adapted for) studies of the dynamics of more than one excited state in the same supercell?*

Currently the time evolution of a single excess charge carrier can be simulated. In principle, the method could be extended to multiple electrons or holes with the accuracy depending on the level of theory used to account for electron-electron or hole-hole interactions. TOF measurements and related electronic measurements are, to the best of our knowledge, done in dilute carrier concentration, 10^{14} - 10^{17} cm⁻³. For comparison, there are about 10^{21} molecules cm⁻³ in typical OS. Hence, carrier-carrier interactions should be negligible and the single excess charge carrier model valid for the purpose of comparing mobilities to experiment.

7) *“CT” is very commonly used to refer to charge transfer in molecular semiconductors rather than charge transport. Is it possible to avoid needing to use the acronym “CT”??*

“CT” has now been replaced by **charge transport** in the whole manuscript.

8) *Figure 4b, although interesting, is not referred to in the main text.*

We have now added a sentence on page 8 to refer to Figure 5b in the revised manuscript: **“We find also a good correlation between our IPR and the localization length L_{τ}^2 divided by the area per molecule within the herringbone layer as shown in Fig. 5b.”**

Reviewer 3:

The manuscript by Giannini and collaborators tackles an important open problem in the field of organic electronics: understanding the underlying principles of hole conduction in molecular organic semiconductors (OS). This problem has been a true head-scratcher for the community. There are a plethora of possible theories that can be applied for modeling hole conductivity. However, each one of them has shown fallacies in predicting one or multiple experimental observations. For the past 5-10 years it has been clear that OS sample a very rich grey area of dynamics mechanisms. Thus, clarity is dearly needed.

This manuscript delivers the needed clarifications. Using known descriptors (e.g., λ and V), the authors describe in minute detail the interesting and multifaceted dynamics that occurs in the initial steps of hole transfer when the initial state is a hole localized on a simple molecule. The authors describe a dynamics that is reminiscent of certain types of biological charge transfer even though there are some clear distinguishing features compared to bio systems (such as 3-dimensionality). I am convinced that this is a quality manuscript that has the chance of being cited hundreds of times, and to become the next “talk of the town”.

In doing my due diligence, I find several aspects, especially of the electronic structure/dynamics methodology that can be explained more in depth:

1) In the interest of a more self-contained publication, the authors should describe their method for generating the position-dependent electronic Hamiltonian matrix. And they should clearly point out any possible shortcoming of the method. This will help other researchers determine what can be improved in the method and whether the predictions can be further crystallized by employing more accurate electronic structure methods.

The calculation of the site energies (diagonal elements) of the position-dependent electronic Hamiltonian was described in the “Methods” section under “Molecular Model”. Here we have augmented this section on page 11 and 12 by including more detail on the sensitivity of the DFT calculations on functional and basis set - as also requested by Reviewer 2. A challenge (rather than a shortcoming) is the inclusion of electronic polarization effects in the calculation of the site energies, in particular when the crystal is formed of polar or hydrogen-bonded molecules (which is not the case in the current systems studied). We have added a comment in this regard on page 12: “We expect that this is no longer a good approximation for crystals formed of polar or hydrogen bonded molecules. In this case the full electrostatics including electronic polarization of the molecules should be included as it is well known that thermal fluctuations of the site energies and therefore reorganization free energies are overestimated for fixed point-charge models.\cite{Seidel09,Oberhofer10acie,Moens10}.” The calculations of the electronic couplings (off-diagonal elements) of the position dependent Hamiltonian was also described in detail in the “Methods” section under “Molecular Model”. We augmented this section on page 12 to comment on the accuracy of FODFT calculations – as requested also by Referee 2. Details on the parametrization of FODFT couplings, i.e. the AOM method, were given in the “Molecular Model” section as well. We further expanded the description of AOM on page 13 by giving the explicit equation of the overlap S_{kl} that is used to estimate couplings at each nuclear time step: “For π -conjugated systems, it is usually sufficient to include only one optimized Slater p -orbital per atom contributing to π -conjugation, in this case: $\bar{S}_{kl} = \sum_i^{\text{atoms}} \sum_j^{\text{atoms}} c_{p\pi,i} c_{p\pi,j} \langle p_{\pi,i} | p_{\pi,j} \rangle$, where i and j run over all π -conjugated atoms in molecules k and l , respectively, $p_{\pi,i}$ is the Slater type orbital p on atom i , and $c_{p\pi,i}$ is the corresponding expansion coefficient obtained by projection on the DFT molecular frontier orbital.” Then we point out possible shortcomings on page 14: “Possible shortcomings of AOM arise from the fact that, although atomic orbitals comprising the HOMO (LUMO) follow the motion of the atoms during the dynamics, the expansion coefficients in Eq.~\ref{eq:Sk} are frozen otherwise. However, our checks indicated that this is a very good approximation, especially for rigid molecules, where orbitals are stable against intermolecular vibrations. More sophisticated interpolation schemes, or machine learning techniques could be used in future to improve reconstruction of the orbitals along the dynamics. Another source of inaccuracy could be the minimal basis set employed in Eq.~\ref{eq:Sk}, where only a single orbital per atom is considered. Although the validity of this approximation has been successfully tested before,\cite{Gajdos14} one could use a larger basis set to improve the orbital representation.”

2) The authors should mention that they have not carried out a study of the initial state dependence. There are conjectures that the initial state of a model should somehow mimic the one realized by experiments. Clearly that is another daunting task. But I wonder if the authors can shed some light on this, or perhaps simply acknowledge this.

We investigated the dependence of mobility and IPR with respect to the chosen initial state, page 17 of the original manuscript version, but this detail was probably difficult to find. We have added a separate paragraph on page 18: “**Dependence of mobility and IPR on the initial state.** As mentioned before, the system was equilibrated with molecule i in the charged state and all other molecules in the neutral state

and the charge carrier wavefunction was initialized accordingly as the frontier orbital localized on molecule i , $|\Psi(0)\rangle = |\phi_i(0)\rangle$. We also investigated other initializations of the system, e.g. starting from configurations where all molecules were initially equilibrated in the charge neutral state, and the charge carrier wavefunction is initially localized on a randomly chosen single molecule m , $|\Psi(0)\rangle = |\phi_m(0)\rangle$. While, obviously, the short-time dynamics differs in each case, after about 200 fs all of the differently initialized systems relax to the same polaronic state with the same IPR and mobility. This is an important observation because it shows that for calculation of charge mobility it is not necessary to reproduce the (usually unknown) initial state in experiment.”

3) *Figure 2 can be improved. I understand that the authors are trying to convey the most information with the least space. I find the figure hard to interpret and perhaps unnecessarily complex.*

We have split Figure 2 in two separate figures, Figure 2 and 3 in the revised manuscript. Figure 2 shows the short time dynamics of polaron motion in atomistic detail (100 fs) and Figure 3 the dynamics on the longer, picosecond time scale in coarse resolution. In new Figure 2 we also introduced color coding to link more clearly the IPR in panel a (g) with the carrier wavefunctions shown in panels b-f (h-l) at the same time slice. We have slightly restructured the labels of the two figures as well and we have modified the main text describing Figure 2 on page 4 and added a few additional lines describing Figure 3 on page 5. We hope the information contained in Figures 2 and 3 is now better presented.

4) *A typical method to include modulation of the Hamiltonian matrix elements with nuclear motion is to model them by an average value and a random distribution of variations from the average with a width that is typically either guessed, or evaluated by MD. If such an approach is employed with appropriate parameters extracted from the authors' simulations (say, from Table S1), would it recover the same dynamics and conclusions? Can the authors make a comment along these lines in the manuscript?*

We assume the reviewer refers to an approach where the time-dependent electronic Schrodinger equation for the charge carrier is solved for time-independent electronic Hamiltonian with the matrix elements for the latter randomly taken from a distribution as described, followed by averaging over all wavefunction evolutions. We think that equilibrium properties such as average size of the polaron, which is related to the delocalization of the Boltzmann-weighted eigenstates, could be fairly well reproduced. However, the approach described would lack two important dynamical ingredients (included in FOB-SH): non-adiabatic coupling between electronic states, which requires explicit time propagation of nuclear degrees of freedom, and the feedback from the electronic to the nuclear dynamics. The former is particularly important in driving the polaron across the material through temporary delocalization in energetically higher-lying electronic states (peaks in IPR values). Hence, we believe that the carrier dynamics and dynamical properties like charge mobility may not be well described with such an approach.

REVIEWERS' COMMENTS:

Reviewer #1 (Remarks to the Author):

The authors have addressed my comments thoroughly. The revised manuscript is improved and I recommend publication with no further changes. The work presents significant results.

S. Skourtis

Reviewer #3 (Remarks to the Author):

The authors have done a good job answering the reviewer's comments, including mine the ones of the other 2 reviewers.

In regards to my comment #2, I think the authors should consider initial states that are superpositions of molecular states. While their answer clearly addresses the initial states of the nuclear dynamics. The electronic initial state is always taken to be a state with a hole localized on a single molecule. Unless there is a strong and clear argument that superposition initial states are unlikely, I would think this needs to be addressed.

I am not suggesting that the authors re-run their simulations with a different initial state, but perhaps can estimate this dependence (if any).

Reviewer 1:

The authors have addressed my comments thoroughly. The revised manuscript is improved and I recommend publication with no further changes. The work presents significant results.

S. Skourtis

Reviewer 3:

The authors have done a good job answering the reviewer's comments, including mine the ones of the other 2 reviewers.

In regards to my comment #2, I think the authors should consider initial states that are superpositions of molecular states. While their answer clearly addresses the initial states of the nuclear dynamics. The electronic initial state is always taken to be a state with a hole localized on a single molecule. Unless there is a strong and clear argument that superposition initial states are unlikely, I would think this needs to be addressed.

I am not suggesting that the authors re-run their simulations with a different initial state, but perhaps can estimate this dependence (if any).

We understand the reviewer's point and we have now addressed this further remark in "Dependence of mobility and IPR on the initial state" pag 19, after mentioning all other initializations tested. "In addition, we notice that detailed balance in FOB-SH algorithm (see Supplementary Figure~1) ensures that even when the electronic carrier wavefunction is initialized as a superposition of localized states (for example an eigenstate of the Hamiltonian, namely $\Psi(0) = \psi_n(0)$), after the aforementioned relaxation period the charge carrier reaches the same equilibrium configuration and follows the same dynamics as in the case of the initially localized charge." Since we are interested in mobility in the diffusive regime, and IPR after electronic relaxation (typically after 200 fs in our simulations), there is no dependence on the particular initialization of the wavefunction as we already mentioned in the last sentence of "Dependence of mobility and IPR on the initial state" pag 19. We also point out that the particular initialization of the wavefunction from a given adiabatic eigenstate of the Hamiltonian (i.e. a superposition of localized states) was also investigated by us in "Carof, A., Giannini, S. & Blumberger, J. J. Chem. Phys. 147, 214113 (2017)." in the context of modeling equilibrium properties of an organic semiconductor model.